# Imprint of urbanization on snow precipitation over the continental USA

Kaustubh Anil Salvi[1] & Mukesh Kumar [1] ✉

Urbanization can alter the local climate through modifications in land-atmosphere feedback. However, a continental scale evaluation of its influence on precipitation phase remains unknown. Here, we assess the difference in the likelihood of snow dominated events (SDEs) over 7,415 urban and surrounding non-urban (buffer) regions across the continental United States. Among 4,856 urban-buffer pairs that received at least five SDEs per year, 81% of urban regions are characterized by a smaller snow probability, 99% by a lower frequency of SDEs, and 57% by faster declining trends in SDEs compared to their buffer counterparts. Notably, urban (buffer) regions with lower snow probability are often characterized by higher net incoming and sensible energy fluxes as compared to buffer (urban) regions, thus highlighting the influence of land-energy feedback on precipitation phase. Results highlight a clear imprint of urbanization on precipitation phase and underscore the need to consider these influences while projecting hydro-meteorological risks.

Snow precipitation supports water resources for more than 1.5 billion people[1]. Alarmingly, climate warming is reducing the fraction of precipitation falling as snow[2,3]. Such a transition is altering water budget partitioning[4], flood risks[5,6], and water supply[7]. In addition, changes in snow vs. rain frequency and/or amount have the potential to also impact the timing and magnitude of seasonal and diurnal peak flows[8,9], summer baseflows[10], evapotranspiration[11], nutrient cycling[12,13], land–atmosphere feedback[14], soil erosion[15], vegetation responses[16], orographic precipitation generation[17], and a host of other coupled ecosystem processes and functions[18,19]. Especially within the urban regions, the phase of precipitation also influences traffic mobility[20] and road accident risks[21,22], frequency and intensity of deicing salt application and its consequent impacts on water quality[23], contamination from runoff[24,25], and vulnerability from extreme hydro-meteorological events[26]. While the influence of climate variations and changes on the likelihood of snow vs. rain has been studied extensively[2,3,27,28], whether, to what extent, and where urbanization impacts precipitation phase over the continental USA (CONUS) remains unknown. Notably, an exclusive focus on evaluating alterations in precipitation phase solely due to global warming, without considering the potential regional impacts of urbanization, would lead to deficient or ineffective projections of its consequences and subsequent mitigation strategies. Given the wide-ranging consequences of precipitation phase, several

of which are noted above, it is imperative to examine the role of urbanization on precipitation phase in order to support long-term water resource planning, ensuring sustainability of ecosystem services, and infrastructure risk assessment and design under changing climate and land cover.

Urbanization's impacts on surface and air temperature due to the so-called urban heat island (UHI) effect[29–32] has been studied extensively, although a majority of these studies have focused on warm/summer periods when snow precipitation is negligible. Furthermore, most past studies on UHI effect did not specifically assess its impact on temperature on the days receiving precipitation, especially with high likelihood of snow. A few studies on the evaluation of the role of urbanization on precipitation phase indicate that urbanization may decrease snow precipitation[33–35] because of UHI. Notably, most of these studies have been conducted at local to regional scale. Over the continent, urbanization's impacts on precipitation phase may however experience significant spatial heterogeneity due to the fact that several locations actually experience urban cooling, instead of heating[36,37]. In fact, the Global Urban Heat Island Data Set, v1 (2013)[38], which consists of data pertaining to differences in average day time maximum and night time minimum temperature during summer months of 2013 for 31,500 urban and corresponding buffer regions reveal that 8949 (28%) urban regions during day time and 11,514 (36%) regions during night

[1]Department of Civil Construction & Environmental Engineering, University of Alabama, Tuscaloosa, AL, USA. ✉e-mail: mkumar4@eng.ua.edu

are cooler than their buffer counterparts. In addition, given urbanization is also known to alter precipitation temporality and its characteristics[39–41], its eventual impact on snow frequency and amount is likely to be influenced by changes in air temperature as well as the temporal distribution and frequency of precipitation events.

This study assesses the differences in modeled average snow probability (SP) between urban areas and their buffer regions, henceforth termed urban–buffer pairs (UBPs), throughout the CONUS. Also evaluated are the differences in the amount of annual precipitation delivered by precipitation events with high snow likelihood. To this end, snow-dominated events (SDEs), i.e., events with likelihood of snow being higher than 50% and 75%, henceforth referred as SDE50 and SDE75, respectively, are identified. We also assess the contrasts in land–atmosphere energy fluxes in urban and buffer regions for each UBP to tease out its potential role on snow precipitation differences between the two regions. Finally, we evaluate the trends in the frequency of SDE50 and SDE75, as well as the snow probabilities associated with them, in all selected UBPs. Unless stated otherwise, reported snow probabilities, their contrasts, and subsequent analysis all throughout this study are based on modeled estimates of likelihood of snow precipitation. More details regarding the selection of UBPs and the data used in this analysis, approach to derive event SP for UBPs, and metrics and thresholds used to define SDEs and assess the differences in their characteristics between urban and buffer regions are presented in "Methods."

## Results

### Average SP contrast between urban and buffer regions

Comparison of the average of modeled SP of SDEs between urban and corresponding buffer regions reveals that in about 81% of UBPs (3944 out of 4856), differences in average SP (daSP, see "Methods" for more details) of SDE50s between buffer and urban regions are positive (Fig. 1). In other words, an overwhelming fraction of buffer regions

encounter SDEs with higher average SP as compared to their urban counterparts. Relatively high magnitudes of positive and negative daSP are existent mainly in the western part of CONUS, while some also lie in the northeastern US as well. UBPs with negative daSP, though in a small fraction, are spread across the CONUS with greater prevalence in the northern Great Plains region. There are 1387 UBPs (out of 4856) with statistically significant daSP ($p < 0.05$) based on $z$-score test. A very high percentage (91%) of these UBPs have positive daSP (Fig. S4). Similar dominance of buffer regions with relatively high SP is noted in case of SDE75 as well, where 79% (3578 out of 4553) of UBPs show positive daSP (Fig. S4) and this percentage increases to 84% (851 out of 1013) for UBPs with statistically significant daSP (Fig. S4). Notably, the spatial pattern (areas with relatively high differences and regions with negative daSP) are consistent irrespective of SP thresholds (50% or 75%) for SDEs.

The SP estimates presented above are based on Dai formula (see "Methods: Derivation of event SP for UBPs," for more details), which is based on air temperature. Evaluation based on Jennings formula, which considers the role of both temperature and relative humidity (RH) on SP, also shows similar results. For example, 86% (84%) of UBPs exhibit positive daSP for SDE50 (SDE75) when calculated using the Jennings' formula, while this magnitude is 81% (79%) in case of Dai's formula (see Figs. 1 and S4). Notably, the daSP obtained using the two approaches exhibit strong linear association (Fig. S4b, c). Considering that both the methods yield very similar results, and the Jennings method requires additional data of RH at 4 km × 4 km resolution and hence possibly the uncertainties inherent with it, only Dai's formula is used for the ensuing analyses.

### Factors controlling spatial distribution and magnitude of urban-buffer SP contrast

To understand the spatial distribution of daSP, first the variation of daSP with difference between mean temperatures of buffer and urban

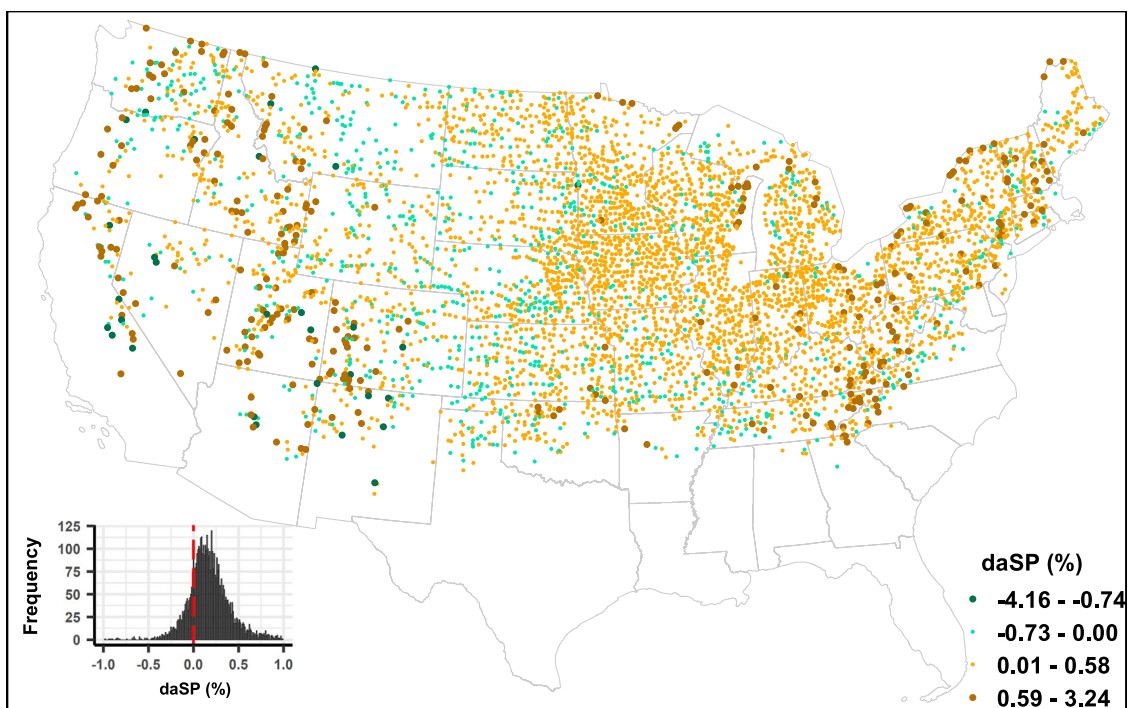

**Fig. 1 | Average snow probability contrast between urban and buffer regions over the continental USA.** Spatial distribution of the difference between average snow probability (daSP), obtained by subtracting the mean snow probability of snow-dominated precipitation events with snow probability >50% (SDE50) in the urban area from that in the corresponding buffer region for all urban–buffer pairs (UBPs), across the continental USA (CONUS). In all, 81% (3944) of the selected UBPs (4856) show positive daSP (inset plot), indicating that the average snow probabilities of SDE50 for those buffer regions are higher than that of their urban counterparts. Source data are provided in Source_data.xlsx, on the sheet called Main_figure_1.

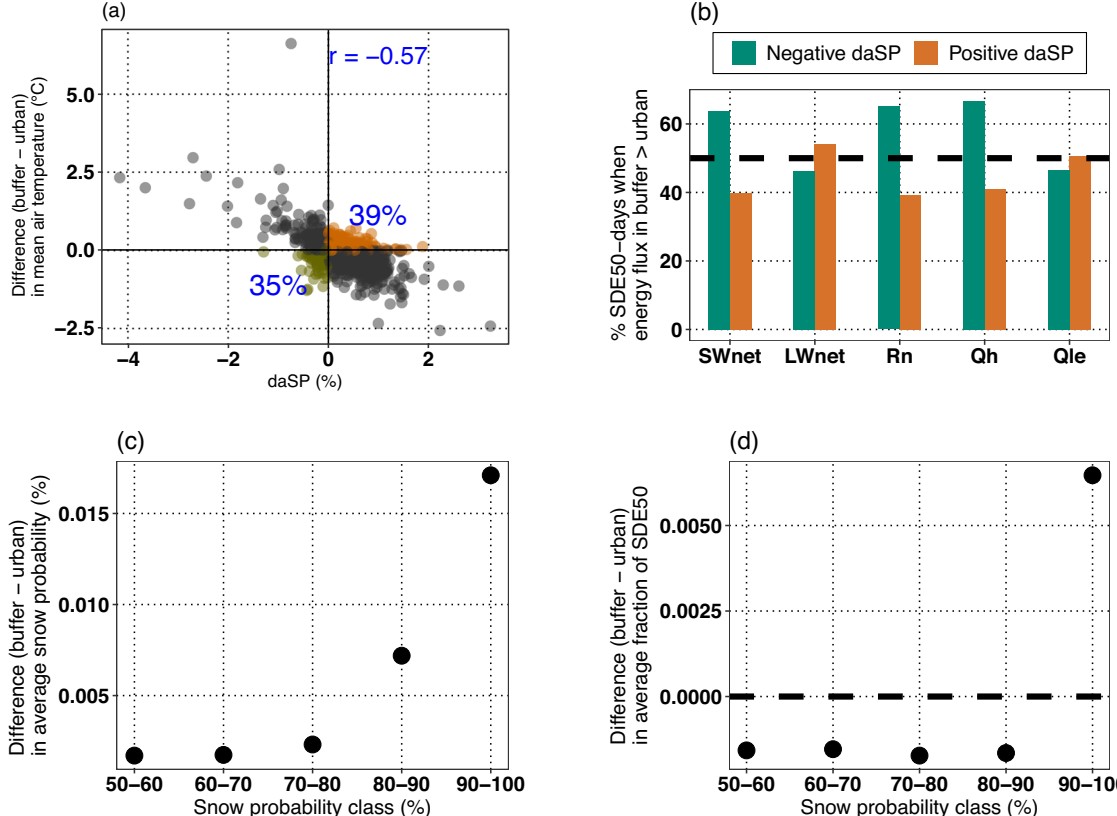

**Fig. 2 | Factors influencing the snow probability contrast between buffer and urban regions.** Contrasts are evaluated for snow-dominated precipitation events (SDEs) with snow probability >50% (SDE50). **a** Difference in mean temperature between buffer and urban regions. Percentage, shown in blue text, in the first (third) quadrant represents the fraction of urban–buffer pairs (UBPs) with positive (negative) difference between average snow probabilities (daSP) with higher average buffer (urban) temperature than in the urban (buffer) region. **b** Average percentage (%) of SDE50-days on which the concerned energy flux is larger in buffer

as compared to urban region. Difference in **c** average snow probability and **d** average fractional occurrence frequency, of SDEs within defined snow probability ranges. The fractional occurrence frequency for an urban (buffer) region is calculated as the ratio of SDEs within a given snow probability range to the total number of SDEs. The average of these ratios over all urban (buffer) regions gives the average fractional occurrence frequency. Source data are provided in Source_data.xlsx, on the sheet called Main figure_2.

regions (Fig. 2a) during 1980–2020 is evaluated. Mean temperatures for urban and buffer regions are respectively obtained by averaging all three-hourly air temperatures during 1980–2020. The correlation between daSP for SDE50 and the difference in mean temperatures between the two regions is found to be $r = -0.57$. In other words, cooler buffer regions w.r.t. their urban counterparts generally experience higher average SP. Similar analysis carried out with difference in mean critical temperature (see "Methods: Mean critical temperature for more details"), which captures thresholded mean air temperature during cooler periods only, indicates a stronger association ($r = -0.68$; Fig. S4a). Absolute value of correlation increases from 0.68 to 0.9 when critical temperatures only during the precipitation events (i.e., mean critical event temperature) are used (Fig. S8b). Considering the SPs calculated here (Fig. S8b) are based on air temperature (Fig. S3), high correlation evident in Fig. S8b is expected. Nonetheless, larger correlations between daSP and difference between mean critical temperatures (Fig. S8a) and daSP and mean event critical temperatures (Fig. S8b) w.r.t. daSP and mean temperature (Fig. 2a) highlight that only a limited fraction of spatial distribution in daSP can be explained based on the difference in mean temperatures between urban and buffer regions, as its spatial variability is markedly different than that of temperature during the SDEs.

Previous literature[32] indicates that difference in air temperature between urban and buffer regions can be due to contrasts in land–atmosphere energy feedback. As shown above, given that the distinctions in daSP between urban and buffer regions are more

related to differences of air temperature during SDEs between the two regions rather than overall difference in average temperatures (Figs. 2a and S8), next we assess the differences in land–atmosphere energy exchange only for the days on which SDE50 occurs in either urban or buffer area (hereafter referred to as SDE50-days) of an UBP (see "Data of UBPs and climate" and "Assessment of differences in land–atmosphere energy fluxes between urban and buffer regions" in "Methods" for more details). For each UBP, we first evaluate the percentage of SDE50-days a given energy component is higher in the buffer region compared to the urban. For each energy component, the averages of these percentages over the selected number of UBPs are obtained next. For UBPs where average SP of SDEs is higher in buffer regions (i.e., daSP is positive), SWnet, Rn, and Qh are all greater in buffer regions on an average of 39.78%, 39.67%, and 40.98% of the days, respectively (Fig. 2b and Table S1, Part A). In other words, these energy fluxes are lower in buffer on a majority of SDE50-days. Mean difference (buffer – urban) in the magnitude of these fluxes across all UBPs and SDE50-days also indicate that, on average, these energy fluxes are of lower magnitude (Fig. S9) in buffer regions with higher SP relative to their urban counterparts. In contrast, LWnet is greater in buffer regions on 54.07% of the SDE50-days and Qle is greater in 50.7% of the days. Given the spatial proximity of urban and buffer regions within an UBP, lower SWnet and Rn in buffer regions on a majority of SDE50-days is largely attributable to higher buffer albedo. In fact, 83.2% and 80.83% of SDE50-days with SWnet and Rn being lower in the buffer region also have albedo higher in them. Similar results are noted

for the UBPs with negative daSP (Fig. 2b and Table S1, Part B). For example, overall, buffer regions with negative daSP experience higher SWnet, Rn, and Qh on 63.68%, 65.08%, and 66.73% of the SDE50-days, respectively. More than 85% of these days are characterized by higher urban albedo (Table S1, Part B).

Overall, the results reveal that the region (urban/buffer) with relatively high average SP generally experiences lower SWnet, Rn, and Qh, and higher land surface albedo. A higher albedo can decrease net incoming radiation by reducing SWnet. While LWnet increases due to concomitant decrease in surface temperature which in turn reduces the outgoing longwave radiation, but given that change in LWnet is generally smaller than SWnet, Rn still is lower with higher albedo. The decrease in Rn and surface temperature consequently diminishes sensible heat feedback[42] to the atmosphere. However, the extent of impact is also determined by several other mediating factors[32], including land roughness properties, emissivity, moisture sources, etc. Since Qle approximately equals Rn minus Qh, and as both Rn and Qh are lower in regions with higher SP, Qle shows no marked contrasts between the two regions (Figs. 2b and S9), with it being higher on only around 51% of the SDE50-days across the UBPs. Overall, considering that Qh has a strong influence on air temperature[43], the contrasts in it between urban and its buffer counterpart contribute to differences in SP. In fact, among the UBPs with higher average SP in buffer regions, i.e., UBPs with positive daSP, 78.64%, 78.27%, and 78.11% of the days with lower Qh in buffer have lower Rn and SWnet, and higher albedo, respectively (Table S2, Part A). Corresponding fractions of 80.28%, 79.59%, and 85.01% for UBPs with higher average SP for urban regions further attest the findings (Table S2, Part B).

It remains unclear whether the increased albedo that leads to reduced Rn and Qh, subsequently resulting in lower air temperatures and an increased likelihood of snowfall, is primarily due to differences in snow cover conditions or land cover characteristics between the two regions. To assess this, we next evaluate the energy fluxes for SDE50-days but only when both urban and their buffer counterparts have snow-free ground (see "Methods" for the details about snow-free ground conditions). For UBPs with positive and statistically significant daSP (i.e., in about 81% of UBPs with statistically significant daSP), albedo in urban is lower for snowcover-free days (Table S1, Part C). However, for UBPs with negative daSP, the fraction of days when buffer Rn and Qh fluxes are lower than urban counterparts are reduced for snowcover-free conditions (Table S1, Part D). Similar evaluations of energy contrasts for snow-covered days (Fig. S10 and Table S3) further reconfirm the underlying role of albedo differences between urban and buffer regions on SP differences. These results highlight that albedo disparities, both during snowcover-free and snow-covered days, are closely associated with disparities in Rn and Qh between urban and buffer counterparts. The overall association between positive daSP and disparities in energy fluxes between urban and rural regions should not be construed as implying that every SDE occurrence will experience these associations. This is particularly true as other dynamic factors such as land surface properties and moisture sources, may also play a mediating role in determining the discrepancies of energy fluxes between urban and rural regions. Another reason is that energy fluxes at daily scale can only be considered as an approximate descriptor of air temperature during SDEs, especially if a day receives multiple SDEs. To assess the imprint of buffer land covers on daSP contrasts on SDE50 days with snowcover-free ground conditions, we stratify the aerial fraction of 11 land covers: water, developed areas, deciduous forest, evergreen forest, mixed forest, shrub (open), grassland, hay, cultivated crops, woody wetlands, and herbaceous wetlands. Fractions of each land cover are first obtained by mapping National Land Cover Database land cover to each buffer region with statistically significant daSP (Fig. S4). The average of these fractions for each land cover is then evaluated separately over UBPs with positive and negative daSP. Results (Table S4) indicate that UBPs with negative daSP are distinguished by a higher forest coverage (deciduous, mixed, and evergreen forests). In contrast, UBPs with positive daSP have a higher proportion of cultivated crops. This imprint is also reflected in sparse occurrence of negative daSP UBPs in croplands of mid-western US. While forest (cultivated crop) land covers usually are characterized by lower (higher) albedos than in built-up areas, it is to be noted that land–atmosphere energy feedback in buffers are also determined by differences in surface and aerodynamic conductances[44] as well. Moreover, the impacts of additional land covers, though individually minor, could wield a more substantial influence than the predominant land cover in the area. Mapping the influences of each land cover on different energy components with desired certainty currently remains beyond the scope of this study.

Additional analysis is carried out to assess the characteristics of SDEs that result in positive daSP over the continent. To this end, first the SDE50 falling within specific SP ranges are identified in both urban and buffer regions. Next, the difference in average SP for each of these ranges is evaluated for each UBP. Average of these differences (Fig. 2c) over all UBPs for different SP ranges indicates that the largest difference in SP is for events falling within the 90–100% SP range. In other words, buffer regions generally have much higher SP than urban regions for SDEs with SP of 90–100%. For other SP ranges, the differences are relatively much smaller. To further confirm that SDEs with SP in the range 90–100% are the main driver of positive daSP overall, we next assess whether these events have at least similar likelihood of occurrences as other SP ranges. To this end, we evaluate the ratio of SDEs with SP falling within a specified range to the total number of SDE50 in each urban or buffer region. Next, the difference of these ratios between buffer and urban regions of each UBP is calculated. Average of these differences over all UBPs (Fig. 2d) shows that for the SP interval 90–100%, the fractions of SDEs between buffer and urban have the higher difference. These results illustrate the predominant influence of SDEs with SP of 90–100% on why the average SP is generally higher for buffer region. Results for SDE75 agree with those for SDE50 (Fig. S11).

## Is there an urban–buffer contrast in precipitation amount delivered by SDEs?

Although an overwhelming fraction of UBPs reveal higher SP in buffer regions, only 54% of the buffer regions, out of 4856 UBPs, receive a higher amount of annual average precipitation delivered by SDE50 (Fig. 3a). This percentage reduces to 52% when urban and buffer regions are compared on the basis of annual average percentage of precipitation received through SDE50 (Fig. 3b). Notably, 99% of buffer regions experience a higher frequency of annual average SDEs (SDE50) (Fig. 3c). Despite this, the reason an overwhelming number of buffer regions do not encounter higher amount of precipitation (or even higher fraction of snow to total precipitation) than their urban counterparts is because for 97% of the UBPs, the buffer regions receive SDE50 at lower intensities as compared to their corresponding urban regions (Fig. 3d). In other words, more number but smaller intensity SDEs result in balancing out of the total precipitation delivered by SDEs between urban and buffer regions. Notably, a smaller intensity of precipitation events, though not for SDEs, in buffer regions w.r.t. urban areas has also been reported previously[45]. Evaluations for SDE75 also show similar results (Fig. S12).

## Differences in temporal trends of SDEs between urban and buffer regions and their controls

The trends of annual frequency of SDEs and annual average SP of SDEs are evaluated next (Fig. 4). In case of annual frequency of SDE50 (Fig. 4a), ~74% (3578 out of 4856) UBPs have statistically significant (s.s.) trends ($p < 0.05$) in both urban and buffer regions. ~98% of these UBPs with s.s. trends (3500 out of 3578) exhibit a declining behavior, i.e., the frequency of annual SDE50 decreases over time in these UBPs.

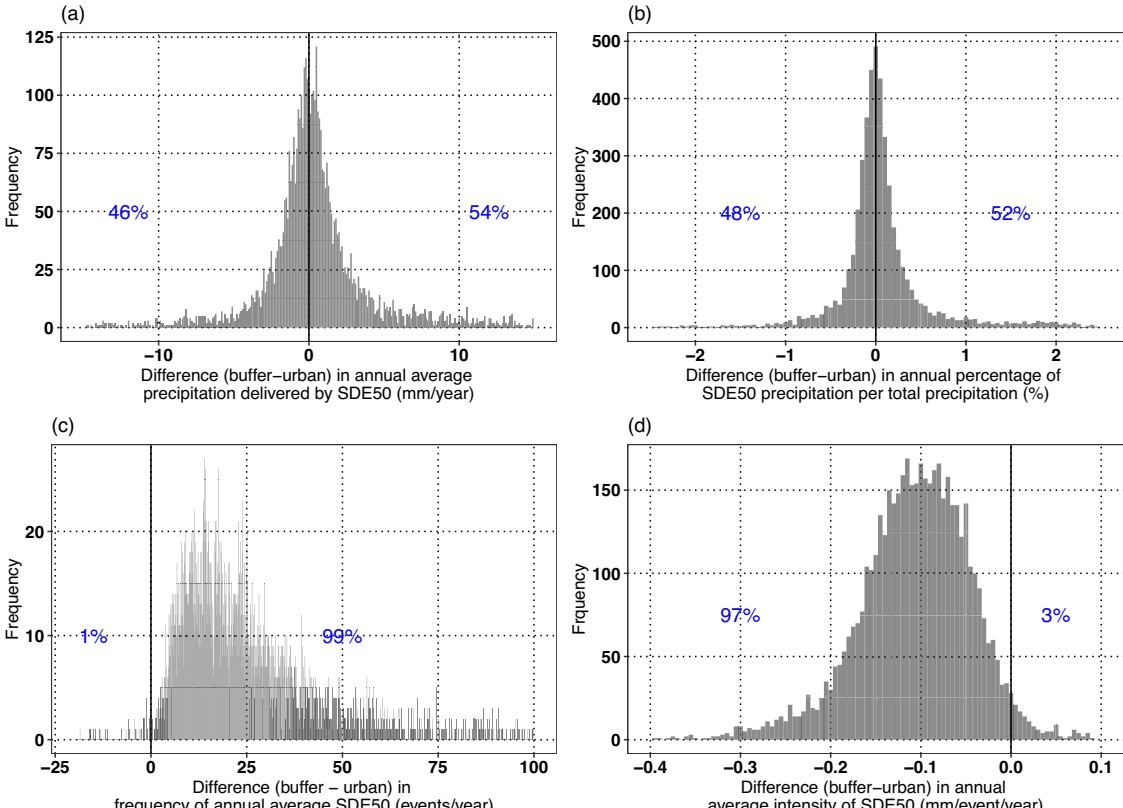

**Fig. 3 | Comparison between urban and buffer regions based on long-term statistical properties of encountered snow-dominated precipitation events.** Frequency distribution of differences (buffer − urban) in **a** annual average precipitation delivered by snow-dominated precipitation events with snow probability >50% (SDE50), **b** annual average percentage (%) of precipitation that is delivered by SDE50, **c** annual average frequency of encountered SDE50, and **d** annual average intensities of precipitation delivered by SDE50, for the selected 4856 UBPs. Notably, an overwhelming percentage (99%) of buffer regions encounter higher frequency of SDE50 but the average intensity of these events is smaller in 97% of them. Source data are provided in Source_data.xlsx, on the sheet called Main_figure_3.

Notably, in ~57% of UBPs with s.s. negative trend (1991 out of 3500), urban regions show a more pronounced negative trend in annual frequency of SDE50 compared to buffer. Trend analysis for SDE75 reveals similar findings (Fig. S13a). While these UBPs are distributed all across the CONUS, those with largest discrepancy in trends between urban and buffers either lie in the northernmost states east of the Mississippi River or are nestled in the mountainous regions (e.g., the Rockies of UT and CO, or the Sierras of California) of the western US (Fig. S14). The more pronounced negative trend in SDEs frequency in urban areas, compared to buffers, may result from two factors: (1) a higher temperature increase in urban regions, which can potentially transition SDEs into rain events, reducing their frequency, and (2) dwindled precipitation events in urban areas, which include a reduction in SDE frequency as well. To validate these possibilities and identify the dominating factor affecting the evolution of annual frequency of SDEs, trends of mean annual temperature and annual frequency of precipitation events are obtained for the UBPs with urban regions showing pronounced negative trends (Table S5, Parts A and B). Among the 1991 UBPs where annual frequency of SDE50 exhibits a more negative trend than buffer, a noteworthy 96% of them (1908 out of 1991) show statistically significant negative trends for annual frequency of precipitation in both urban and buffer regions, with the urban regions showing a more substantial negative trend (Table S5, Part A). In contrast, only 40% of the 1991 UBPs have higher magnitude of annual average temperature trends for urban regions. A similar pattern is noticed while examining trends for SDE75 (Table S5, Part B). These results indicate that for urban regions with higher negative trend of SDE frequency, the primary factor for the expressed contrasts is the disparity in the trends of annual precipitation frequency, rather than the air temperature.

For annual average SP, a smaller fraction (~20%) of UBPs show s.s. trends (982 out of 4856) in both urban and buffer regions. However, 98% of these UBPs (958 out of 982) exhibit declining trends in both urban and buffer regions. In 70% of these UBPs (675 out of 958), urban regions show a more pronounced negative trend than buffer areas. Trend analysis for SDE75 reveals similar findings (Fig. S13b). Largest differences in trend magnitude are noted in central and northwestern US (Fig. S14c, d). Next, we assess the potential impacts of differing trends in air temperature and critical event temperature, the two variables that show a significant association with the difference in average SP between urban and buffer regions (Figs. 2a and S8), on the varied trends in average SP. The analysis reveals that, more so than the trend in annual average temperature, higher positive trends in annual average event critical temperature are closely associated with a more rapid decline in the annual average SP of SDEs (Table S5, Parts C and D).

## Discussion

The study presents a continental-scale analysis of the influence of urbanization on precipitation phase carried out over the CONUS. The results indicate that in a significant fraction of UBPs, urbanization has led to an overall reduction in the frequency of snow dominated events and also the modeled probability of snowfall during these events. Moreover, the reduced frequency of SDEs in urban areas intensifies over time for a majority fraction of UBPs. Even though buffer regions generally receive a higher number of snow events each year, no marked contrast between urban and buffer counterparts is found for the amount of precipitation received as snow. This is largely because most of the urban areas receive snow precipitation from higher intensity events. Difference between average temperature of the

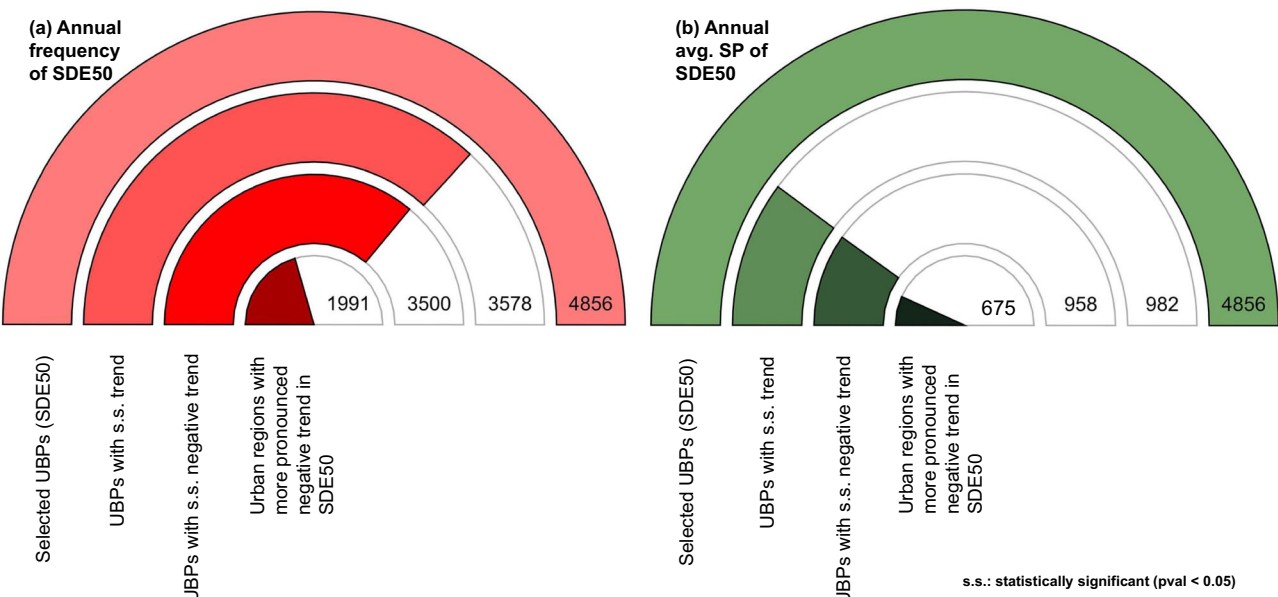

**Fig. 4 | Contrasts in temporal variations of frequency and average snow probability of snow-dominated precipitation events over urban and buffer regions.** Contrasts are evaluated for snow-dominated precipitation events (SDEs) with snow probability >50% (SDE50). Variables considered include **a** annual frequency of SDE50, and **b** annual average snow probability (SP) for SDE50. "s.s. trend" denotes statistically significant (Pval <0.05). In both the cases, a dominant percentage of UBPs (out of those with s.s. trends) exhibit negative trends (3500 out of 3578 (**a**) and 958 out of 982 (**b**)) where urban regions illustrate a more pronounced negative trend as compared to their buffer counterparts (1991 out of 3500 (**a**) and 675 out of 985 (**b**)). Source data are provided in Source_data.xlsx, on the sheet called Main_figure_4_and_Table_S5.

buffer and urban region is found to modulate the spatial distribution of SP contrasts. However, results also indicate that differences in event temperatures between buffer and urban regions, which have a more direct control on SP contrast, can be quite different than differences in average temperature. While delineating all the controls on event temperature contrasts between urban and buffer regions is out of scope of this study, our analysis shows that differences in net short-wave radiation, net radiation, and sensible heat, driven in part by albedo contrasts, are associated with divergences in temperature and consequently SP. Overall, the results also show that average SP is higher for buffer regions due to a higher fraction of events with SP >90% in the buffer regions.

While utmost care has been taken regarding the choice of data used and the methodology implemented to ensure uncertainties are kept to a minimum, they inevitably do exist. For example, North American Land Data Assimilation System (NLDAS) data of precipitation and temperature, while used extensively, has uncertainties inherent in it as it is a post-processed reanalysis data. Alternative is to use the station-level data of occurrence of snow vs. rain that have been used to develop SP models[46,47]. However, these point observation data suffers from several limitations, including (1) lack of sufficient stations to adequately perform a continental analysis over the selected UBPs, (2) lack of cotemporaneous, long-term data (e.g., 67% of stations residing within UBPs have data for <10 years) even in the UBPs that have data of precipitation phase in either urban or the buffer region, (3) lack of representativeness to the neighboring areas. Considering these limitations, the NLDAS data emerges as a preferred choice for the current application. It is to be acknowledged that coarse NLDAS data (spatial resolution of 0.125° × 0.125°) of albedo and land–atmosphere energy exchanges that are sourced from NLDAS project make a comprehensive attribution analysis of SP contrasts a challenge. Although albedo data can be obtained at higher resolutions through remote sensing, these estimates are unavailable on days with SDE50 due to cloud cover. Daily remotely sensed albedo data products for CONUS usually only offer the daily albedo estimate within a specific timeframe or window, excluding data for the cloudy days. Notably, comparison of modeled trend of annual event average SP from NLDAS data with observations, at 121 locations where the modeled trends are statistically significant (p value <0.05), reveals that the direction of trends (i.e., positive or negative) align at 67% of the stations (Fig. S15). Furthermore, among the 17 UBPs with at least one observation station in both urban and buffer regions, and where the SP contrast is statistically significant, 71% of the UBPs exhibit alignment between modeled and observed regarding whether the urban or the buffer region has a higher SP (Fig. S15). These findings indicate that evaluations based on NLDAS data generally align with observations. Discrepancies in the SP or its contrasts based on the two data sources likely arise from (1) the scale mismatch, where observational data at a single point may lack representativeness of neighboring areas; (2) the inherent uncertainties in the two datasets.

The temperature threshold used here, which is based on the hyperbolic tangent function, to assess the precipitation phase has been found to vary across the northern hemisphere[47]. Notably, our main conclusions remain valid even while considering SDEs with varied SP (50% and 75%) or in other words for different temperature thresholds. It is to be acknowledged that there could be other confounding factors such as anthropogenic heat sources in urban areas[48], and differences in urban features, their spatial distribution, and properties such as roughness characteristics, moisture content, and emissivity, which could also contribute to contrasts in snow probabilities between urban and buffer regions. However, lack of detailed data, especially within-urban regions at the continental scale, and uncertainty in current land surface models precludes such analysis. Complexity in analysis emanating from transient dynamics and heterogeneity of snow and land cover, in both urban and buffer regions, is another major limitation of this study. For example, removal of snow, alteration of land cover properties, heterogeneity introduced by faster or slower snowmelt in sheltered vs. exposed areas, and spatial variation of surface temperature, roughness properties, wind fields, etc., within both buffer and urban regions, can all influence land–atmosphere energy exchange and consequently the characteristics of SDEs. As more data becomes available, future studies may consider these details to provide a more refined understanding of their impacts.

Considering that alterations in snow vs. rain precipitation can have wide-ranging impacts, such as on water supply[1], flood risk[6] and its quantification[26], traffic operations and safety[21,22], water quality[23], and drought and its prediction[49], the results of this study emphasize the importance of considering the influence of urbanization on snow precipitation characteristics while projecting influences on the aforementioned variables. Importantly, failing to consider the potential influences of urbanization on the characteristics of snow precipitation could lead to incomplete or ineffective projections of the hydro-meteorological risks it presents. The findings of this study have substantial significance, particularly for urban planners, policymakers, and researchers, as they may aid in proactive assessment and mitigation to negate the adverse impacts of impending urban expansion on snow patterns and related system functions. Future research that focuses on examining the impact of the extent, spatial configuration, and other properties of built structures, including their rate of change, on snow precipitation characteristics in different UBPs, is expected to enhance these assessments and mitigation designs. Additional insights may also be gained by assessing the role of specific land covers in the buffer regions on snow dynamics, and eventually on snow-probability contrasts. The acquisition of novel, high-resolution spatiotemporal data pertaining to land surface and snow cover characteristics is likely to play a pivotal role in this regard.

## Methods

Subsequent sections provide comprehensive information pertaining to the data and the adopted procedure for evaluating SP contrasts between urban and corresponding buffer regions (Fig. S1). This includes details regarding delineating buffer regions around each urban area, downscaling and spatially explicit mapping of climate variables, obtaining spatially-averaged representative temperature and precipitation time series for both urban and buffer regions, and finally obtaining differences between average snow probabilities for SDEs (SP >50% and 75%). Additionally, these sections elaborate on the metrics and thresholds employed for the evaluation of the differences in snow precipitation characteristics between buffer and urban regions.

### Data of UBPs and climate

In this study, the Global Rural-Urban Mapping Project, Version 1 (GRUMPv1): Urban Extent Polygons, Revision 02 is used to obtain the data of urban regions[38]. GRUMPv1 provides data of 75,445 urban areas across entire globe out of which 9359 urban regions are situated within the CONUS (Fig. S1a). The data have been widely used in studies to map and understand UHIs[50–52]. Corresponding to each urban area, surrounding non-urban or buffer zone is mapped using the Buffer function in ArcGIS Desktop 10.6.1 (Fig. S1b). Buffer corresponding to each of the urban area is drawn using a 10 km distance threshold from the boundary of demarcated urban areas. It is to be noted that while different approaches exist for the selection of size of buffer regions, including those based on distance thresholds (e.g., use of 1 km threshold[37] and 10 km threshold[32]), fraction of urban area-based approach[53,54] (e.g., area of buffer region between 50% and 150% of urban area), and pixel-based approach[55], the choice of distance threshold used here is guided by the need to feasibly and representatively ascertain SP contrasts within UBPs over the entire CONUS. Use of 10 km threshold allows ample samples from the buffer region. Increasing the width of buffer region beyond 10 km engulfs surrounding urban regions while reducing the width to say 1 km is not advisable as the downscaled climate variables are at 4 km resolution. Notably, the 10 km threshold has also been selected for analysis in other previous studies[32,38] related to UHI effect as well.

For ensuing analysis, precipitation and air temperature data from the NLDAS project[56] are used. These data are available at 0.125° spatial and an hourly temporal resolution, and have been widely used in a range of disciplines such as hydrology[57,58], UHI impact on public health[59,60] and agriculture[61]. Although other data at much finer spatial resolution exists, they are however generally at a temporally coarser resolution. The sub-daily resolution of the temperature and precipitation time series in the NLDAS dataset is crucial for appropriate apportioning of precipitation into snow and rain[62] and also for evaluation of SP[46] for a given event.

To evaluate the energy exchange between land and atmosphere on the days when SDE50s are encountered by either of the two regions in an UBP (SDE50-days), outputs from the Noah land surface model simulations, which are also obtained for the NLDAS project[56], are used. The Noah model serves as the land component in several coupled land–atmosphere modeling systems, including the Weather Research and Forecasting (WRF) regional atmospheric model, the NOAA-NCEP coupled Climate Forecast System (CFS), and the Global Forecast System. The energy components that are considered for this analysis include (1) net shortwave radiation flux (SWnet), (2) net longwave radiation flux (LWnet), (3) net radiation flux (Rn), (4) sensible heat flux (Qh), and (5) latent heat flux (Qle). Net radiation flux is calculated as the sum of net shortwave and net longwave radiation fluxes from the Noah model output. Data of surface albedo from the model output are also used.

### Derivation of event SP for UBPs

The procedure to estimate the probability of precipitation event being snow involves multiple steps (Fig. S1). These include:

**Climate data mapping onto 4 km × 4 km grid over the CONUS.** Considering that more than 80% of the 9359 urban regions situated within the CONUS, based on GRUMPv1 data, have areas less than the spatial extent of a NLDAS grid (area ~0.125° × 0.125° or ~190 km² at equator), the coarse resolution temperature data are downscaled to 4-km grid using inverse distance bilinear interpolation approach. For obtaining interpolated temperature at any given 4 km × 4 km grid center, the distances between the center of this grid and centers of four surrounding NLDAS grids are used as weights. Similar spatial downscaling of temperature data has also been performed in other data products[63].

Temporally, temperature data are up-scaled to 3-h resolution. This is because the parameters of SP function used in this study (explained later in the "Methods: SP in urban and buffer regions" section) were established using data at 3-hourly resolution. Precipitation data from NLDAS are, however, transferred as-is to the encompassing 4 km × 4 km resolution grids, in part because precipitation generally shows significant spatial heterogeneity governed by additional meteorological, physiographic, and topographic attributes[64–66]. Adopting a simple interpolation scheme for precipitation data may introduce large uncertainties leading to erroneous findings[56]. Notably, also within the reanalysis datasets[67], spatial distribution of precipitation are generally determined by the model outputs or spatially-explicit radar based observations, instead of using simple (e.g., bilinear) spatial interpolations of the observations.

**Obtaining spatially averaged climate of urban and buffer areas.** For each UBP, all 4 km × 4 km grids with more than 50% of their areas within the urban or buffer regions, respectively, are identified (see Fig. S1c). Out of 9359 UBPs (Fig. S2a) within the CONUS, 7415 UBPs (Fig. S2b) have at least one 4 km × 4 km grid each within urban and buffer regions. For the ensuing analyses, these 7415 UBPs are selected. For computational tractability, a single temperature and precipitation time series are obtained by spatially averaging data of selected grids in each of the 7415 urban and buffer regions. For example, an urban region consisting of n number of grids (at 4 km resolution) with more than 50% of their areas within the region, temperature time series for the region is generated by averaging the temperature time series for *n* grids.

**SP in urban and buffer regions.** Phase of precipitation at the ground surface is influenced by the properties of the atmosphere through which it travels. Different models for precipitation phase determination[68] have been proposed in previous studies, including those based on (1) air temperature and RH thresholds[46,69,70] and (2) atmospheric models with microphysics schemes that track a hydrometeor from its formation[71]. Here we obtain SP based on a hyperbolic tangent function (Fig. S3) whose parameters have been established using 3-hourly weather reports data[46]. The function (Eq. 1) has been used in numerous studies to perform snow vs. rain partitioning[27,47]. Hereinafter, this function will be referred as Dai formula.

$$SP = -48.2292 \times (\tanh(0.7205 \times (temperature - 1.1662)) - 1.0223) \quad (1)$$

SP is also evaluated using a bivariate logistic regression model[47] (termed as Jennings formula henceforth) that links SP with both RH and temperature

$$SP = 100/(1 + \exp(\alpha + \beta T_s + \gamma RH)) \quad (2)$$

where $\alpha = -10.04$, $\beta = 1.41$, $\gamma = 0.09$. For this, the RH is calculated based on Clausius–Clapeyron equation at the spatial and temporal resolutions of NLDAS data using Eq. 3.

$$RH = 0.263 \times specific\ humidity \times pressure / \exp((17.67 \times (T_s - 273.16))/(T_s - 29.65)) \quad (3)$$

where specific humidity is expressed in kg/kg, pressure in Pascal, and Ts in Kelvin. Specific humidity and pressure data are obtained from NLDAS datasets. RH obtained in previous step is spatially downscaled (at 4 km resolution) and temporally up-scaled at 3-h resolution using the same approach as that for the temperature (see "Climate data mapping onto 4 km × 4 km grid over the CONUS" in "Methods").

## Metrics and thresholds to assess the differences in snow precipitation characteristics between urban and buffer regions

As the premise of this study is to evaluate the impacts of urbanization on snow events, here we focus on SDEs, i.e., events for which the SP ≥50%, i.e., the probability of event being in solid phase is higher or equal to than it being in the liquid phase. We also define the metrics and variables that allow the assessment of contrasts in snow characteristics between urban and buffer regions. Finally, UBPs where robust statistical analyses of the variables under consideration can be performed are identified.

**Thresholds used to define SDEs.** Here two threshold snow probabilities are used to define SDEs. These include SDE with SP greater than 50% and 75%. These events are termed as SDE50 and SDE75, respectively. These snow probabilities correspond to air temperature (Tthresh) = 1.146 °C and 0.341 °C, respectively (Fig. S3), based on the Dai formula.

**Metrics used to assess snow contrasts between urban and buffer regions.** Six metrics are used to assess the impact of urbanization on snow characteristics. The first metric used here is the difference (buffer−urban) in average snow probabilities for SDEs, henceforth called the daSP, between the buffer and the urban regions during 1980–2020 (Fig. S1d). This metric helps identify whether the buffer or the urban region receive SDEs with a higher average probability of snow. The second metric quantifies the difference (buffer−urban) in the annual average amount of precipitation delivered by SDEs during the analysis period, i.e., 1980–2020. The third metric evaluates the difference (buffer − urban) between the annual average percentages of precipitation that is delivered by SDEs. The fourth metric compares the intensities of precipitation (=precipitation amount/number of SDEs) delivered by SDEs. It is quantified in mm/events/year by subtracting

the average annual intensity of SDEs over the urban region from that in the buffer. The final two metrics compare the temporal trends of (1) annual frequency of SDEs and (2) annual average SP of SDEs (for both SDE50 and SDE75) for each UBP. The trends are calculated using the non-parametric Sen Slope estimator.

**Mean critical temperature.** Mean critical temperature and mean critical event temperature are evaluated to assess the role of air temperature during cooler periods on SP. Mean critical temperature for urban/buffer region is obtained by only averaging the air temperature values below 1.146 °C during 1980–2020, after substituting the temperature data below −4 °C with −4 °C. The reason for using 1.146 °C as an upper threshold is that this temperature corresponds to 50% SP (Fig. S3). As SP below −4 °C remains almost constant, the temperature values below this threshold are replaced by −4 °C. The selection of these thresholds enables evaluation of a temperature metric, which can be easily calculated and is expected to capture the variations of SP across regions. Mean critical event temperature is calculated in a similar manner as the mean critical temperature. However, temperature values only during the precipitation events are considered for its evaluation.

**Selection of UBPs.** Even though the SPs are obtained for all 7415 UBPs, all evaluations in this study are performed only for the UBPs that receive at least 5 SDEs per year in both urban and buffer regions. Out of 7415 UBPs (see Fig. S2b), 4856 (4553) UBPs encounter at least 5 SDE50 (SDE75) per year (Fig. S2c, d) in both urban and buffer regions of a UBP. This additional criterion is imposed to ensure sufficient sample size (frequency of SDEs) while obtaining the metrics.

## Assessment of differences in land–atmosphere energy fluxes between urban and buffer regions

Spatially averaged time series of energy components and albedo for each UBP are obtained using the same procedure as temperature as discussed above in the sections "Climate data mapping onto 4 km × 4 km grid over the CONUS" and "Obtaining spatially averaged climate of urban and buffer areas." These data are obtained for a 41-year period (1980–2020).

Contrasts in energy fluxes are evaluated for 1387 UBPs with statistically significant daSP (based on 41 years of SP calculations for SDE50s (see supplementary Fig. S4)). For each of these UBPs, spatially and temporally averaged daytime (i.e., hours when net shortwave radiation flux >0) energy fluxes such as SWnet, LWnet, Rn, Qh, Qle, and albedo are calculated individually for both urban and buffer regions. The evaluations are performed for all days receiving SDE50 in either urban or buffer region. As antecedent states at sunrise and cumulative Qh to the atmosphere drives the net change in boundary layer thickness and air temperature[43] during a day, daily energy fluxes between urban and buffers are compared. Evaluations are also performed separately for a subset of these days when (1) the ground is snowcover-free and (2) ground is covered with snow, for both the urban and buffer regions. Snowcover-free days based on NLDAS data are defined as the days when modeled albedo of both the regions is <0.35 and snow-covered days are identified by complimentary magnitude of albedo (≥0.35). The threshold is conservative given that the upper limit of all land covers considered in the NLDAS project has an albedo of 0.33.

## Data availability

NLDAS gridded precipitation, temperature, energy fluxes, and albedo hourly data can be downloaded from https://disc.gsfc.nasa.gov/datasets?keywords=NLDAS. The Global Rural-Urban Mapping Project, Version 1 (GRUMPv1): Urban Extent Polygons, Revision 02 can be retrieved from https://sedac.ciesin.columbia.edu/data/set/grump-v1-urban-ext-polygons-rev02/data-download. Data used for the analysis

of snow characteristics contrast between urban and buffer regions may be accessed from https://zenodo.org/records/10723791. Source data are provided with this paper.

## Code availability

The codes to generate figures may be accessed from https://zenodo.org/records/10723791. Detailed information about required input files and steps to run the codes are provided in readme.docx.

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

## Acknowledgements

We acknowledge partial financial support provided by NOAA (NA22NWS4320003), NSF (DGE-2152140), and DOE-SCIDAC for this research (funding acquisition: M.K.).

## Author contributions

M.K. conceived the idea, designed the problem, and acquired the funding. M.K. and K.S. formulated the methodology. K.S. worked on data acquisition, software selection, code writing, visualizations, and basic validations. Formal analysis and investigations were conducted by M.K. and K.S. K.S. wrote the original draft, which was reviewed, modified, and edited by M.K. M.K. handled project administration and supervision.

## Competing interests

The authors declare no competing interests.
