## [Peer Review File · Nature Communications]

Imprint of Urbanization on Snow Precipitation over the Continental USAREVIEWER COMMENTS

Reviewer #1 (Remarks to the Author):

The manuscript explored the differences in the snow dominated events(SDEs) between urban and surrounding buffer regions across the continental United States. The main conclusion was that the urbanization has a negative impact on the occurrence probability of snow events, indicating that urban regions tend to have smaller snow probability, lower frequency, and decline in SDEs. These findings are interesting. However, the following issues should be clarified.

Scientific relevance:

The major issue is that the current manuscript does not provide a very good scientific question, and it suffers from limited physical explanations, and an overall lack of discussions as how this study builds upon the large body of studies around the world.

- (1)Do the authors consider the changes of urban regions during urbanization process?
- (2)Could the authors explain what does positive daSP(in line 91) and other negative mean? Could please give a clear explanation?
- (3)Is it reasonable to using a constant 10km distance threshold for all the urban regions?
- (4)The methodology, especially the detailed information of the climate data used should be furthered clarified.
- (5)As the authors mentioned, there are still some uncertainties in the manuscript. Therefore, how did the authors do their best to minimum?
- (6)The first sentence in line 216 was too absolute. Did the authors review all the previous literature?
- (7)It is strongly recommended to proofread the paper thoroughly again.

Reviewer #2 (Remarks to the Author):

The manuscript documents differences between urban and adjacent buffer zones in snow to precipitation ratio, snow frequency, snow intensity, and trends in these metrics. The noteworthy results include that a) buffer areas tend to receive more frequent but less intense snowfall events that are characterized by a higher snowfall fraction relative to urban areas and b) trends in the frequency of snow events and their snow to rain ratio are more negative in urban areas than buffer areas.

The work is original and the claims are supported by the evidence. However, the broader impacts of the work and its significance are relatively weak, at least as currently framed in the paper. The authors mention why precipitation phase matters, but stronger motivation for looking at effects of urbanization specifically on precipitation phase would improve the impact of the paper. While I believe the work is absolutely worth publishing, I am not sure the significance is great enough for publication in Nature Communications.

I had several questions about the methods, mostly clarifications of text that I found unclear. My only major concern with the methods, based on my current understanding, is the spatial resolution and downscaling of the climate data. I am concerned that inverse distance weighting downscaling of ~12km data to 4km does not capture the real spatial heterogeneity of climate fields at 4km, rather it just smooths a coarse grid. I was concerned that this would make it difficult to capture contrasts between urban-buffer pairs, but the analysis demonstrated contrasts, so perhaps this is less of a problem than I initially thought. Addressing this would require using a different climate dataset (and I understand 4km hourly data over CONUS is a big ask) or a more sophisticated downscaling method. The latter might be worth considering.

Overall, the paper is thorough, interesting, and mostly well presented.

Sincerely,

Abby Lute

A few less important but general comments:

I'm not super familiar with Nature Communications formatting. Is it normal to have such lengthy pointers to the methods section in the main text and to have so much content in the figure captions?

It would be very helpful (and more pleasant to look at) if the color coding of positive and negative daSP in the figures and tables throughout the paper was consistent.

I highlighted some grammatical errors, but there were many that I didn't highlight. Feedback from a science writer or editor would improve the readability of the paper.

Below I provide more specific feedback.

Abstract

Line 16- 'the' not needed

Line 16- The fact that it is unknown is a relatively weak justification for a study. Why do we want to know it? Why does it matter if precipitation phase depends on urbanization? Answering these questions might provide a stronger motivation for the study and get the reader more interested.

I don't think you need so many numbers in the abstract, it makes it a little harder to get at the big picture. Especially in Line 19, these numbers could be omitted for readability.

Line 19- delete 'an'

Line 20- could 'relatively rapid decline' be rephrased to clarify that we are talking about long-term historical trends here?

Line 22- Suggestion: replace 'are found to be usually' with 'are often'

Lines 21-24- is this lower of snow probability in general or relative to their counterpart (urban/buffer)? It is unclear at the moment. And it seems like 'fraction' could be left out.

Lines 24-26- Rephrase. Precipitation phase is not susceptible to urbanization, it is influenced by it though. It might be fairer to say 'Results show that trends in snow frequency and snow to rain ratio are more negative in urban areas than buffer areas, highlighting the effects of urbanization on precipitation phase.'

Main

Line 29- resource should be plural

Line 32- have not has

Line 37- suggestion: replace 'does the urbanization impact' with 'urbanization impacts'

Lines 35-38- again this is very weak motivation for the study. Why do we care about the effect of urbanization specifically on precipitation phase?

Line 42- replace 'may result in decrease in' with 'may decrease'

Lines 50-51- This sentence is unclear and seems overly wordy. Is the intention: "The extent/intensity of the UHI effect has previously been mapped for summer and winter days."?

Line 52- 54- check grammar in this sentence

Line 59- delete 'as'

Lines 65-66 Just saying trends would be clearer than temporal evolution.

Average snow probability contrasts between urban and buffer regions

Figure 1- Do I understand correctly that a daSP of 0.5 for example means that there is less than 1% difference in snow probability between urban and buffer?

Figure S7- please add 1:1 line on panel b) and c). Even though it is mostly obvious, would be nice to add a panel like a) but for the Dai SP, just to make it easy to compare the two methods.

Factors controlling spatial distribution and magnitude of urban-buffer snow probability contrast

Line 102- is 'mean temperature' the mean annual temperature? Mean temperature during SDEs? Please provide a little more description. It is hard to interpret the findings presented in this paragraph without having a little better understanding of the metric.

Figure S8- average event critical temperature should be defined in the methods section. The gridlines are inconsistent in different areas of the plot, which seems odd.

Line 111- replace Figure references with metric references. E.g. "larger correlations with average critical temperature and average event critical temperature compared to mean temperature highlight...'

Otherwise the reader has to translate what the figures say in order to make sense of the sentence.

Figure 2b- This is pretty challenging to interpret as there is a lot of translation that has to occur. It's hard to suggest alternatives without seeing the data, but I wonder if a bar chart showing the mean differences in each flux (W/m²) on SDE50 days would work. For each flux there would be two bars- one for positive daSP and one for negative daSP. I think it would help to keep the calculation the same between positive and negative daSP (i.e. either urban – buffer or buffer – urban, not switching like you're doing now). This would have the added advantage of showing the relative magnitudes of the differences in the different

fluxes. Also, Q_{le} is not defined in the text.

Q_{le} presents different behaviour than the other fluxes (Figure 2b) but this divergent result is not discussed in the text. How is this behaviour explained?

Line 151- I don't think these other factors were noted above.

Line 157- Why not use energy fluxes during the events instead of for the full day? Maybe this was in the methods and I missed it.

Figures 2c and d- please remove grey space so that plot content can be enlarged and easier to see. Might be helpful to label the colorbars to indicate what metric they are showing. I prefer a color bar that diverges at 0 for a metric like this. As it is, the colors suggest that green and brown cells mean different things, when in reality they might both be showing positive values. Why are some heat map cells divided by a black line and others aren't? Same comments apply to Figure S9.

Lines 160-161- by 'difference in SP' do you mean the same daSP that's been discussed throughout the paper? If not, how is this different? What is the SP for a difference in SP? For example, if this is daSP then is the SP associated with it the urban SP or the buffer SP?

Line 162- what is the difference in fractions of SDEs? What is the fraction that is referred to? This whole paragraph should be better explained in the methods. For example, the grouping into seven bins should be in the methods.

Line 165- what are frequency-fractions? Where was this defined?

Lines 159-171- I read this paragraph many times and could not figure out what it said. There seem to be several metric definitions missing, or maybe they are defined in the methods but the same terminology is not used here. I was unable to evaluate the science in this paragraph because I couldn't understand the presentation of it.

Is there an urban-buffer contrast in precipitation amount delivered by snow-dominated events as well?

Line 174- delete 'as well'

Line 175- I would rephrase the first few sentences of this paragraph to say that there is little contrast in a) the amount of precipitation delivered or b) the precipitation fraction between urban and buffer, instead of focusing on the relatively small differences you found.

Figure 3b- if this is a difference between two fractions that each range from 0 to 1, how can there be values greater than 1 or less than -1?

Figure 3- again the grid lines are inconsistent. Not a huge deal, just annoying. In the caption, SD50 should be changed to SDE50 and either iSDE needs to be changed to match the abbreviation in the methods section (ipptSDEs) or vice versa.

Line 187- was there any explanation for the difference in intensity offered by this previous work?

Differences in temporal trends of snow-dominated events between urban and buffer regions, and their controls

Line 192- delete both instances of 'or'

Lines 195-197- This sentence is mostly talking about afSDE and Figure 4a, but then it says 'both afSDE and aaSPSDE decrease with time in these UBPs'. This sentence makes sense with Figure 4a, but contradicts Figure 4b.

Figure S11- The clustering in these plots seems very strange. Is this because only statistically significant

trends are shown? If so, the fact that only these trends are shown should be noted in the caption. Even if that is the case, I don't understand why there would be clusters at the 0 temperature trend lines. Trend units should be noted on x and y axis labels or in caption.

Figure 4- the description for the smallest semicircle in a) and b) is unclear. What does 'relatively rapid' mean? How about something like 'trend in urban region is more negative than buffer region' or 'urban trend is more negative than buffer trend'? Same applies to Figure S12.

Can you comment on the spatial distribution of the s.s. trends and of the locations where urban trends are greater than buffer trends?

Conclusions

Line 216- should be 'analysis' not 'analyses'

Line 218- rephrase as 'urbanization decreases snow probability'. 'negative impact' sounds like a value judgement

Line 218- 'Alarmingly' seems inappropriate here without being explicit about why this should be alarming. Also, why would this be? Is it because of interactions between urbanization and changing climate, because characteristics of urban areas are changing, or something else?

Lines 224-239- I understand the issues of scope and data limitations, but it seems like one could comment more specifically on potential contributions to urban-buffer differences. For example, it seems likely that buffer regions have more vegetation and soil surfaces as opposed to concrete/asphalt surfaces and this could affect local convection, moisture availability, and latent heat flux in specific ways.

What are the implications of less snow in urban areas? In many ways this seems like a good thing- it would likely contribute to less congestion, fewer accidents, and less funding needed to be spent on road clearing and repairs. On the other hand, if this is precipitation is coming as rain instead perhaps it increases flood risk and poses challenges to drainage systems in urban areas which generally have surfaces with lower permeability.

Why do buffers tend to get more frequent but less intense SDE's? What are potential explanations for this? This seems like one of the more interesting findings to me.

How do you explain large negative daSP values in the West? Is there something unique about these locations?

Maybe also mention that this study does not assess snow on the ground or how snowfall persists on the landscape, but is focused on snowfall. A nice future work piece might be looking at complementary data sources (ground observations or radar) to see both if observations backup what you see with modeling/reanalysis and also to consider contrasts in snow on the ground / snow persistence.

Methods

Data of UBPs and climate

Figure S1- why are there extra curvilinear lines in the rasterized buffer example? Urban legend label needs to be adjusted to be next to the legend key. The map in a) could use a country outline since the color for the country and the plot background are the same. In b) it looks like there are some exceptions to the 10km buffer, where areas within 10km are excluded. Please explain these in the methods.

A quick glance suggests that the data used for the energy fluxes also contains hourly temperature and

precipitation. Why are these not used instead of the NLDAS data? Also, why use shortwave and net instead of shortwave and longwave?

Derivation of event snow probability (SP) for UBPs

Downscaling of temperature (and energy fluxes) to 4km using an interpolation approach seems like it would not capture urban/buffer contrasts well at all, especially since more than 80% of urban areas considered have areas smaller than the NLDAS grid.

Line 297- the citation referenced here used inverse distance weighting to get a grid from station observations to compare to a downscaled temperature dataset, the approach was not used to downscale temperature per se. Perhaps there is a better citation you can use to justify your approach. How is an event defined? Is it one three-hour time period with snow? This is important to note since in other works snow events have been defined as three days, or other lengths of time.

Metrics and thresholds to assess the differences in snow precipitation characteristics between urban and buffer regions

Line 346- could rephrase as 'events for which the SP is ≥ 0.5 '

Lines 352-353- Do you mean the snow probabilities correspond to these temperatures using the Dai method? Because for the Jennings approach SP would be a function of RH as well and you wouldn't be able to define a single temperature for each SP. Please clarify in the text. Also the caption of Figure S3 should note that this is based on the Dai formula.

Line 354- please include units for each metric

Line 356- please add (buffer – urban) or (urban – buffer) to clarify how daSP is calculated.

Line 360- for pptSDE is this the average precipitation of an event or the annual sum of precipitation from these events?

Line 362- how is the intensity of SDE defined?

Line 364-367- why are the trends evaluated differently? Why not use the same method for both metrics? For the trend metrics are the annual frequency and annual average SP summed across the urban and buffer components of each UBP? Or averaged?

The description of the average critical temperature makes no sense to me. Why are temperature values averaged if they are below 1°C? Are these temperature values at any point in the year or just during SDEs? Why do temperature values below -4°C need to be reset to -4°C? How is this a critical temperature, what does it tell you? Maybe I'm missing something here because this is very unclear to me.

Line 374- is this 5 between the urban and buffer areas, or 5 each?

Assessment of differences in land-atmosphere energy fluxes between urban and buffer regions

Figure S4- Suggestion: I prefer when the color ranges are symmetric around 0. I think the text 'UBPs with positive daSP: ~91%' belongs in the manuscript text, not embedded in the figure. Histogram needs a y axis label. I would also add state outlines below the points to help the viewer better understand the spatial distribution of the points. And you might remove the box around the histogram plot. All the same comments apply to Figure 1, Figure S5, and Figure S6 as well.

Line 386- capitalize figure

Line 389- so this analysis uses days of SDEs, not strictly the 3-hour time periods when they occur? Why not be more specific about the time?

Line 391- replace 'correspond to' with 'are defined as days' to clarify that this isn't fact, it is how you're defining them for the purpose of this study.

Reviewer #3 (Remarks to the Author):

The analysis presented uses temperature and precipitation data from the North American Land Data Assimilation System, combined with a simple statistical model of precipitation type, to evaluate differences in the probability and quantities of snowfall in Urban versus Buffer (adjacent non-urban) areas (termed Urban-Buffer pairs). The authors find that Urban areas, owing to lower land surface albedo and associated greater quantities of net solar radiation, net radiation, and greater sensible heat flux, have lower snowfall probabilities than adjacent buffer areas. The analysis is interesting, well considered, and thorough and the contribution will be valued by many in the scientific community and by the general public. My comments are intended to address a few potential issues inherent in the analysis and text.

1) The authors rely on a model and not observations. I assume there is a good justification for this as it seems obvious to do, but why isn't there an observation-based analysis in the manuscript to complement the modeling results (e.g. using the Dai data set)? Related to this issue, the text should be adapted to insert, in all instances, the word "modeled" when referring to snowfall probabilities and other model-derived quantities.

2) Throughout the paper there seems to be an over-use of abbreviations which make the paper unnecessarily challenging to read - the section starting on line 190 is a prime example.

3) Greater emphasis needs to be placed on the analysis of Snow Dominated Events for situations with pre-existing snow cover versus events without pre-existing snow cover. Wouldn't the presence of snow largely negate land cover associated differences between urban versus buffer areas as everything is covered in highly reflective snow?

4) What is the impact of the background land cover type of the buffer areas? Given that albedo differences between pair members seems to play a large role in determining differences in snowfall probability, it would stand to reason that the albedo of buffer areas would be important. For example, a forest typically has a lower albedo than a grassland. In addition, a small amount of snow cover will dramatically alter the albedo grassland whereas it will have a more subtle impact on the albedo of a forest. These differences in buffer albedo would then have an appreciable impact on the difference between the buffer albedo and an adjacent urban area. Can this be evaluated with the data presented herein? Can the analysis be stratified to look at the differences in snowfall probability across differences in background buffer land cover type? Or at the very least, this effect seems worthy of discussion within the manuscript as the spatial patterns in daSP seem to evoke a land cover control.

5) In the section starting on line 190 related to trends: Can you report the trend in daSP? Is it possible that if warming potentially pushes more pairs toward the rain-snow threshold temperature, might the subtle differences in temperature between pair members cause a greater decrease in snowfall

probability an urban areas relative to the corresponding buffer area? I think the analysis suggests there is greater sensitivity in the urban areas but an explicit analysis of daSP trend seems warranted.

RESPONSE TO THE FIRST REVIEWER'S COMMENTS

R1-1: The manuscript explored the differences in the snow dominated events (SDEs) between urban and surrounding buffer regions across the continental United States. The main conclusion was that urbanization has a negative impact on the occurrence probability of snow events, indicating that urban regions tend to have smaller snow probability, lower frequency, and decline in SDEs. These findings are interesting. However, the following issues should be clarified.

Response: We thank the reviewer for appreciating the novel findings of the paper. We have addressed all the comments. More details are as follows.

R1-2: Scientific relevance: The major issue is that the current manuscript does not provide a very good scientific question, and it suffers from limited physical explanations, and an overall lack of discussions as how this study builds upon the large body of studies around the world.

Response: Thanks for this note.

Precipitation phase, i.e., precipitation falling as snow vs. rain, plays a significant role on water supply, and ecosystem processes and functions, and this is widely recognized (see a succinct summary and references listed in lines 28-34). Possible impacts of precipitation phase specifically on urbanized regions' function and sustainability are now highlighted (lines 34-38). The research gaps associated with evaluation of precipitation phase changes, and the consequences of not considering effects of urbanization (lines 40-47) and are also now described in the text. These influences make the central driving question of this research, that is "whether, to what extent, and where urbanization impacts the precipitation phase over the continental USA" is significantly important. See minor edits in lines 38-40 that further refine the scientific question and the need to address it. The importance of the undertaken problem, and the current research gaps are detailed in the first two paragraphs of the *Main* section. These paragraphs also lay out how this study builds upon a large body of literature (evidenced by 41 references just in these two paragraphs).

Physical explanations of results, specifically aimed at understanding the snow probability contrasts between urban and buffer regions vis-à-vis differences in inherent snow probability regimes and land-atmosphere energy exchange, are presented at length in the section *Factors controlling spatial distribution and magnitude of urban-buffer snow probability contrast* (see lines 105-205).

R1-3: Do the authors consider the changes of urban regions during urbanization process?

Response: Thanks for this perceptive comment. Our analysis does implicitly account for the impacts of changes in urban region on local climate, and consequently snow precipitation characteristics, given that the NLDAS temperature data used here is derived based on NCEP Reanalysis, which assimilates observational data during the simulation period. Notably, observational data will have the response of urbanization already encoded in it.

Explicit analysis of the contribution of changes in urban regions on snow precipitation characteristics has not been performed here. Such a task is challenging currently, if not insurmountable, given the lack of long-term data tracking the changes in urban regions over the entire CONUS during the analysis period. Even if such a data existed, isolating the role of temporally evolving contributions of changing urban regions on snow precipitation characteristics would remain challenging given the concomitant variations/changes in climate and other land covers and their non-linear varied influences on urban climate. One potential strategy to identify this influence would be to use a continental-scale coupled land-atmosphere simulations at appropriately fine enough spatio-temporal resolution. Performing such simulations is expected to be exacting. Following text has been added to the manuscript (Line 312-321)

The findings of this study have substantial significance, particularly for urban planners, policymakers, and researchers, as they may aid in proactive assessment and mitigation to negate the adverse impacts of impending urban expansion on snow patterns and related system functions. Future research that focuses on examining the impact of the extent, spatial configuration, and other properties of built structures, including their rate of change, on snow precipitation characteristics in different UBPs, is expected to enhance these assessments and mitigation designs. Additional insights may also be gained by assessing the role of specific land covers in the buffer regions on snow dynamics, and eventually on snow-probability contrasts. The acquisition of novel, high-resolution spatiotemporal data pertaining to land surface and snow cover characteristics is likely to play a pivotal role in this regard.

R1-4: Could the authors explain what does positive daSP (in line 91) and other negative mean? Could please give a clear explanation?

Response: The first mention of 'daSP' is on line 81. For additional details, we pointed the readers to *Methods* section. There, we have now modified the text and provided additional details regarding the calculation and interpretation of daSP (lines 441-445).

The first metric used here is the difference (buffer – urban) in average snow probabilities for snow dominated events, henceforth called the daSP, between the buffer and the urban regions during 1980-2020 (Figure S1 (d)). This metric helps identify whether the buffer or the urban region receive SDEs with a higher average probability of snow.

Explicit definition of positive/negative daSP are also described in the manuscript (lines 136-139) and in the Figure 1 caption

R1-5: Is it reasonable to using a constant 10km distance threshold for all the urban regions?

Response: We have now clarified the choice of this threshold furthermore (see Lines 342-351). The following text has been added to the manuscript:

It is to be noted that while different approaches exist for the selection of size of buffer regions, including those based on distance thresholds (e.g., use of 1 km threshold (Shashtri et al., 2017) and 10 km threshold (Manoli et al., 2019)), fraction of urban area-based approach (Zhou et al., 2013;Peng et al., 2012) (e.g., area of buffer region between 50 to 150% of urban area), and pixel-based approach (Zhao et al., 2014), the choice of distance threshold used here is guided by the need to feasibly and representatively ascertain snow probability contrasts within UBPs over the entire CONUS. Use of 10 km threshold allows ample samples from the buffer region. Increasing the width of buffer region beyond 10 km is observed to engulf surrounding urban regions and reducing the width to say 1 km is not advisable as the downscaled climate variables are at 4 km resolution. Notably, the 10 km threshold has also been selected for analysis in other previous studies (Manoli et al., 2019; CIESIN, 2016) related to study of UHIs as well.

- 1) Shastri, H., Barik, B., Ghosh, S., Venkataraman, C. and Sadavarte, P., 2017. Flip flop of day-night and summer-winter surface urban heat island intensity in India. Scientific reports, 7(1), pp.1-8.
- 2) Manoli, G., Fatichi, S., Schläpfer, M., Yu, K., Crowther, T.W., Meili, N., Burlando, P., Katul, G.G. and Bou-Zeid, E., 2019. Magnitude of urban heat islands largely explained by climate and population. Nature, 573(7772), pp.55-60.
- 3) Zhou, B., Rybski, D. and Kropp, J.P., 2013. On the statistics of urban heat island intensity. Geophysical research letters, 40(20), pp.5486-5491.
- 4) Peng, S., Piao, S., Ciais, P., Friedlingstein, P., Oettle, C., Bréon, F.M., Nan, H., Zhou, L. and Myneni, R.B., 2012. Surface urban heat island across 419 global big cities. Environmental science & technology, 46(2), pp.696-703.
- 5) Cui, Y., Xu, X., Dong, J. and Qin, Y., 2016. Influence of urbanization factors on surface urban heat island intensity: A comparison of countries at different developmental phases. Sustainability, 8(8), p.706.
- 6) Zhao, L., Lee, X., Smith, R.B. and Oleson, K., 2014. Strong contributions of local background climate to urban heat islands. Nature, 511(7508), pp.216-219.

7) Center for International Earth Science Information Network (CIESIN), Columbia University. Documentation for the Global Urban Heat Island (UHI) Data Set, 2013. Palisades NY: NASA Socioeconomic Data and Applications Center (SEDAC). <http://doi.org/10.7927/H44M92HC>

R1-6: The methodology, especially the detailed information of the climate data used should be further clarified.

Response:

In-depth details about the data used in this study are provided in 'Data of UBPs and climate' in 'Methods' section (lines 334-370). These include information regarding the sources of the data, their spatio-temporal resolutions, and pre-processing that underwent before their use in the analysis.

To further clarify the methodology, edits have been made in lines 325-331 to provide overall summary of the procedure adopted to calculate urban-buffer snow probability contrast. See below:

Subsequent sections provide comprehensive information pertaining to the data and the adopted procedure for evaluating snow probability contrasts between urban and corresponding buffer regions (Figure S1). This includes details regarding delineating buffer regions around each urban area, downscaling and spatially-explicit mapping of climate variables, obtaining spatially-averaged representative temperature and precipitation time series for both urban and buffer regions, and finally obtaining differences between average snow probabilities for snow dominated events (snow probability > 50% and 75%).

Also, Supplementary figure S1 has been modified to include stepwise description of the approach.

R1-7: As the authors mentioned, there are still some uncertainties in the manuscript. Therefore, how did the authors do their best to minimum?

Response: The inherent uncertainties associated with this analysis have been acknowledged (and further expanded) in the manuscript (see Lines 289-299).

For some of the uncertainties, we have carried out additional analysis to understand how a particular uncertainty might affect the results. For example, uncertainty associated with snow probability calculation is addressed by using different approaches (see lines 403-426) and results are shown in supplementary figure S7.

Similarly, we conducted analysis for multiple temperature thresholds to make sure that the reported results are not just valid for one threshold (see lines 436-439).

Despite efforts to exercise due diligence such as by considering appropriate data sets and methodology, many uncertainties inevitably do exist. For example, NLDAS data of precipitation and temperature, which has been used here and also in thousands of other studies, have uncertainties inherent in them. These uncertainties have been duly acknowledged. In fact, in order to check if the uncertainty due to downscaling (bilinear interpolation) approach adopted in this study to obtain 4km spatial resolution data is minimum, analysis is carried out where urban-buffer contrast in temperature is obtained for PRISM data (spatially fine resolution dataset) and results are compared with those obtained with NLDAS dataset (not included in the manuscript). We point the reviewer to the response for point R2-2 for further details.

Jennings, K.S., Winchell, T.S., Livneh, B. and Molotch, N.P., 2018. Spatial variation of the rain-snow temperature threshold across the Northern Hemisphere. *Nature communications*, 9(1), p.1148.

R1-8: The first sentence in line 216 was too absolute. Did the authors review all the previous literature?

Response: We have modified the text in the manuscript as follows (line 265).

To our knowledge, this study presents the first continental scale analyses of the influence of urbanization on precipitation phase carried out over the CONUS.

R1-9: It is strongly recommended to proofread the paper thoroughly again.

Response: Thanks for this note. We have gone through the manuscript thoroughly and have made necessary edits to remove grammatical errors and semantic ambiguities.

RESPONSE TO THE SECOND REVIEWER'S COMMENTS

The manuscript documents differences between urban and adjacent buffer zones in snow to precipitation ratio, snow frequency, snow intensity, and trends in these metrics. The noteworthy results include that a) buffer areas tend to receive more frequent but less intense snowfall events that are characterized by a higher snowfall fraction relative to urban areas and b) trends in the frequency of snow events and their snow to rain ratio are more negative in urban areas than buffer areas.

Response: We thank Dr. Lute for noticing the novelty of results. We have addressed all her comments. More details are as follows.

R2-1: The work is original and the claims are supported by the evidence. However, the broader impacts of the work and its significance are relatively weak, at least as currently framed in the paper. The authors mention why precipitation phase matters, but stronger motivation for looking at effects of urbanization specifically on precipitation phase would improve the impact of the paper. While I believe the work is absolutely worth publishing, I am not sure the significance is great enough for publication in Nature Communications.

Response: We have now added explanations regarding the need to assess the effects of urbanization on precipitation phase (see lines 34-47).

Especially within the urban regions, the phase of precipitation also influences traffic mobility (Liang et al., 1998) and road accident risks (Saha et al., 2016; Salvi et al., 2022), frequency and intensity of deicing salt application and its consequent impacts on water quality (Kaushal et al., 2005), contamination from runoff (Hall et al., 2016; Meyer et al., 2011), and vulnerability from extreme hydro-meteorological events (Yan et al., 2018). While the influence of climate variations and changes on the likelihood of snow vs. rain has been studied extensively (Barnett et al., 2005; Knowles et al., 2006; Klos et al., 2014; Ning et al., 2015), whether, to what extent, and where urbanization impacts precipitation phase over the continental USA (CONUS) remains unknown. Notably, an exclusive focus on evaluating alterations in precipitation phase solely due to global warming, without considering the potential regional impacts of urbanization, would lead to deficient or ineffective projections of its consequences and subsequent mitigation strategies. Given the wide-ranging consequences of precipitation phase, several of which are noted above, it is imperative to examine the role of urbanization on precipitation phase in order to support long-term water resource planning, ensuring sustainability of ecosystem services, and infrastructure risk assessment and design under changing climate and land cover.

- Liang, W. L., Kyte, M., Kitchener, F. & Shannon, P. Effect of environmental factors on driver speed: a case study. *Transportation Research Record* 1635, 155-161 (1998).
- Saha, S., Schramm, P., Nolan, A. & Hess, J. Adverse weather conditions and fatal motor vehicle crashes in the United States, 1994-2012. *Environmental health* 15, 1-9 (2016).
- Salvi, K. A. & Kumar, M. Rainfall-induced hydroplaning risk over road infrastructure of the continental USA. *Plos one* 17, e0272993 (2022).
- Kaushal, S. S. et al. Increased salinization of fresh water in the northeastern United States. *Proceedings of the National Academy of Sciences* 102, 13517-13520 (2005).
- Hall, S. J. et al. Stream nitrogen inputs reflect groundwater across a snowmelt-dominated montane to urban watershed. *Environmental science & technology* 50, 1137-1146 (2016).
- Meyer, T., De Silva, A. O., Spencer, C. & Wania, F. Fate of perfluorinated carboxylates and sulfonates during snowmelt within an urban watershed. *Environmental science & technology* 45, 8113-8119 (2011).
- Yan, H. et al. Next-generation intensity-duration-frequency curves for hydrologic design in snow-dominated environments. *Water Resources Research* 54, 1093-1108 (2018).
- Barnett, T. P., Adam, J. C. & Lettenmaier, D. P. Potential impacts of a warming climate on water availability in snow-dominated regions. *Nature* 438, 303-309 (2005).
- Knowles, N., Dettinger, M. D. & Cayan, D. R. Trends in snowfall versus rainfall in the western United States. *Journal of Climate* 19, 4545-4559 (2006).
- Klos, P. Z., Link, T. E. & Abatzoglou, J. T. Extent of the rain-snow transition zone in the western US under historic and projected climate. *Geophysical Research Letters* 41, 4560-4568 (2014).
- Ning, L. & Bradley, R. S. Snow occurrence changes over the central and eastern United States under future warming scenarios. *Scientific Reports* 5, 1-8 (2015).

R2-2: I had several questions about the methods, mostly clarifications of text that I found unclear. My only major concern with the methods, based on my current understanding, is the spatial resolution and downscaling of the climate data. I am concerned that inverse distance weighting downscaling of ~12km data to 4km does not capture the real spatial heterogeneity of climate fields at 4km, rather it just smooths a coarse grid. I was concerned that this would make it difficult to capture contrasts between urban-buffer pairs, but the analysis demonstrated contrasts, so perhaps this is less of a problem that I initially thought. Addressing this would require using a different climate dataset (and I understand 4km hourly data over CONUS is a big ask) or a more sophisticated downscaling method. The latter might be worth considering.

Response: Thanks for this perceptive comment. To our knowledge, there is no other continental dataset for the analysis period which is at both similar spatial (4 km or finer) and temporal (3-hr or finer) resolution. Downscaling of other datasets that are spatially coarse but temporally fine is likely to experience the same challenge as we face here for the NLDAS.

To check whether there is agreement in the contrasts of air temperature (and consequently snow characteristics) between urban and buffer regions between downscaled NLDAS data and other finer spatial-resolution datasets, as per the reviewer's suggestion, we have carried out an additional analysis using PRISM data (~4km spatial

and daily temporal resolution) after upscaling the 3-hrly NLDAS used in this study to daily resolution, for a 10 year period (2011-2020). This involved:

1. First obtaining PRISM data for urban and buffer areas using the same approach used for NLDAS data (see steps a, b, and c in Figure S1).
2. Upscaling of NLDAS temperature time series for urban and buffer regions from 3-hr to daily scale so that comparison with daily PRISM data can be performed.
3. For the days on which SDE50 events (i.e., precipitation events with likelihood of snow being higher than 50%) occur in either urban or buffer region based on the NLDAS data, differences in temperature between buffer and urban regions are evaluated for both NLDAS and PRISM datasets.
4. If both the datasets identify the same region (say urban) to be warmer or cooler on average, then it is considered they agree in representation of temperature contrasts on snow dominated days.

Result from this analysis is shown below (Illustration 1). It is observed that urban-buffer contrasts based on downscaled NLDAS and PRISM data largely agree with each other, with only less than 4% of urban-buffer pairs showing different contrasts. This indicates that the reported contrasts are also inherent in alternative data, thus eliciting further confidence in our analyses. Since the PRISM data is at daily resolution, we did not evaluate the snow probability contrasts here as equation-1, which is used to evaluate snow probability, is valid for a 3-hourly resolution data.

Illustration 1: Comparison of urban-buffer contrasts in temperature for SDE50-days using PRISM and NLDAS dataset. Comparison could not be performed for 3.7% of the sites because of lack of data (Filled values).

Overall, the paper is thorough, interesting, and mostly well presented.

Response: We thank Dr. Lute for these encouraging words.

R2-3: I'm not super familiar with Nature Communications formatting. Is it normal to have such lengthy pointers to the methods section in the main text and to have so much content in the figure captions?

Response: We checked the authors' guidelines for 'Nature Communication' again. We did not find any instructions particularly for the pointers. We have followed the format used in other papers published in the journal, and are open to changing these pointers as needed per editorial instructions.

In response to the reviewer's note, we have modified the text to minimize mentions of pointers. Regarding figure captions, the maximum length is 350. The figure captions in the manuscript are well within the word limit.

R2-4: It would be very helpful (and more pleasant to look at) if the color coding of positive and negative daSP in the figures and tables throughout the paper was consistent.

Response: Thanks for this suggestion. We have modified the figures in the manuscript to have the same color coding for positive and negative daSP.

R2-5: I highlighted some grammatical errors, but there were many that I didn't highlight. Feedback from a science writer or editor would improve the readability of the paper.

Response: We have carefully gone through the text and eliminated grammatical errors. Some of the examples are (not limited to) ';' added after CONUS (line 86), 'the' added before 'average' (line 79), etc.

Abstract

R2-6: Line 16- 'the' not needed

Response: The text has been modified as per the suggestion (Line 16)

R2-7: Line 16- The fact that it is unknown is a relatively weak justification for a study. Why do we want to know it? Why does it matter if precipitation phase depends on urbanization? Answering these questions might provide a stronger motivation for the study and get the reader more interested.

Response: Thanks for this note. We have made edits in lines 34-47. We also point the reviewer to the response to comment R2-1, above.

R2-8: I don't think you need so many numbers in the abstract, it makes it a little harder to get at the big picture. Especially in Line 19, these numbers could be omitted for readability.

Response: We have made edits to the Abstract to keep it at minimal.

R2-9: Line 19- delete 'an'

Response: The text has been modified in the manuscript (Line 19).

R2-10: Line 20- could 'relatively rapid decline' be rephrased to clarify that we are talking about long-term historical trends here?

Response: We have modified the text in lines 18-21 as follows.

Of the 4,856 urban-buffer pairs that received at least five SDEs per year, this analysis reveals a smaller snow probability, lower frequency, and a faster declining trend in SDEs over time in a majority of urban regions compared to their buffer counterparts.

R2-11: Line 22- Suggestion: replace 'are found to be usually' with 'are often'

Response: Thanks for pointing this out. The text has been modified as per the suggestion (Line 22).

R2-12: Lines 21-24- is this lower of snow probability in general or relative to their counterpart (urban/buffer)? It is unclear at the moment. And it seems like 'fraction' could be left out.

Response: We have now clarified that lower snow probability is relative to their buffer counterpart (See lines 18-21). We have modified the text (lines 21-24) as follows.

Notably, urban (buffer) regions with lower snow probability are often characterized by higher net incoming and sensible energy fluxes as compared to buffer (urban) regions, thus highlighting the influence of land-energy feedback on precipitation phase.

R2-13: Lines 24-26- Rephrase. Precipitation phase is not susceptible to urbanization, it is influenced by it though. It might be fairer to say ‘Results show that trends in snow frequency and snow to rain ratio are more negative in urban areas than buffer areas, highlighting the effects of urbanization on precipitation phase.’

Response: Thanks for pointing this out. The trend related findings are mentioned in lines 18-21. Lines 24-26 are modified as follows.

Results highlight a clear imprint of urbanization on precipitation phase and underscore the need to consider these influences while projecting hydro-meteorological risks.

Main

R2-14: Line 29- resource should be plural

Response: The text has been modified as per the suggestion (Line 28). Thanks for pointing this out.

R2-15: Line 32- have not has

Response: The text has been modified as per the suggestion (Line 31).

R2-16: Line 37- suggestion: replace ‘does the urbanization impact’ with ‘urbanization impacts’

Response: Per reviewer’s suggestion, we have modified the text (Line 39).

R2-17: Lines 35-38- again this is very weak motivation for the study. Why do we care about the effect of urbanization specifically on precipitation phase?

Response: Thanks for pointing this out. We have made edits in lines 34-47. We also point the reviewer to the response to comment R2-1.

R2-18: Line 42- replace 'may result in decrease in' with 'may decrease'

Response: The text has been modified as per the suggestion (Line 52-54).

R2-19: Lines 50-51- This sentence is unclear and seems overly wordy. Is the intention: "The extent/intensity of the UHI effect has previously been mapped for summer and winter days."?

Response: This sentence has been removed from the manuscript.

R2-20: Line 52- 54- check grammar in this sentence

Response: The text in the manuscript has been modified (line 61-64) as follows.

In addition, given urbanization is also known to alter precipitation temporality and its characteristics³⁹⁻⁴¹, its eventual impact on snow frequency and amount is likely to be influenced by changes in air temperature as well as the temporal distribution and frequency of precipitation events.

R2-21: Line 59- delete 'as'

Response: The text has been modified (Line 66).

R2-22: Lines 65-66 Just saying trends would be clearer than temporal evolution.

Response: The text has been modified as per the suggestion (Line 72-73).

Finally, we evaluate the trends in the frequency of SDE50 and SDE75, as well as the snow probabilities associated with them, in all selected UBPs.

R2-23: Average snow probability contrasts between urban and buffer regions

Figure 1- Do I understand correctly that a daSP of 0.5 for example means that there is less than 1% difference in snow probability between urban and buffer?

Response: Yes, that is correct.

R2-24: Figure S7- please add 1:1 line on panel b) and c). Even though it is mostly obvious, would be nice to add a panel like a) but for the Dai SP, just to make it easy to compare the two methods.

Response: Thanks for this suggestion. We have modified figure S7 as per the suggestions. However, a plot such as panel 'a' for Dai's SP could not be generated. This is because the hyperbolic tangent function of snow probability proposed by Dai (2008) is only dependent on temperature. We have already shown that in figure S3.

R2-25: Factors controlling spatial distribution and magnitude of urban-buffer snow probability contrast

Line 102- is 'mean temperature' the mean annual temperature? Mean temperature during SDEs? Please provide a little more description. It is hard to interpret the findings presented in this paragraph without having a little better understanding of the metric.

Response: Thanks for this comment. We have added the following text in the manuscript for clarification (Line 107-110).

To understand the spatial distribution of daSP, first the variation of daSP with difference between mean temperatures of buffer and urban regions (Figure 2(a)) during 1980-2020 is evaluated. Mean temperatures for urban and buffer regions are respectively obtained by averaging all three-hourly air temperatures during 1980-2020.

R2-26: Figure S8- average event critical temperature should be defined in the methods section. The gridlines are inconsistent in different areas of the plot, which seems odd.

Response: The 'Average critical temperature' section is now called "Mean critical temperature" in the manuscript (line 454-464). More details have been provided therein. The inconsistent grid lines in the figure background was a result of figure resolution. It has been fixed now. Thanks for pointing this out.

Mean critical temperature and mean critical event temperature are evaluated to assess the role of air temperature during cooler periods on SP. Mean critical temperature for urban/buffer region is obtained by only averaging the air temperature values below 1.146°C during 1980-2020, after substituting the temperature data below -4°C with -4°C. The reason for using 1.146°C as an upper threshold is that this temperature corresponds to 50% SP (Figure S3). As SP below -4°C remains almost constant, the temperature values below this threshold are replaced by -4°C. The selection of these thresholds enables evaluation of a temperature metric, which can be easily calculated and is expected to capture the variations of SP across regions. Mean critical event temperature is calculated in a similar manner as the mean critical temperature. However, temperature values only during the precipitation events are considered for its evaluation.

R2-27: Line 111- replace Figure references with metric references. E.g. “larger correlations with average critical temperature and average event critical temperature compared to mean temperature highlight...” Otherwise the reader has to translate what the figures say in order to make sense of the sentence.

Response: Thanks for this suggestion. We have modified the text (lines 120-125) and provided the necessary details along with figure references as follows.

Nonetheless, larger correlations between daSP and difference between mean critical temperatures (Figures S8(a)) and daSP and mean event critical temperatures (Figures S8(b)) w.r.t. daSP and mean temperature (Figure 2(a)) highlight that only a limited fraction of spatial distribution in daSP can be explained based on the difference in mean temperatures between urban and buffer regions, as its spatial variability is markedly different than that of temperature during the SDEs.

R2-28: Figure 2b- This is pretty challenging to interpret as there is a lot of translation that has to occur. It's hard to suggest alternatives without seeing the data, but I wonder if a barchart showing the mean differences in each flux (W/m²) on SDE50 days would work. For each flux there would be two bars- one for positive daSP and one for negative daSP. I think it would help to keep the calculation the same between positive and negative daSP (i.e. either urban – buffer or buffer – urban, not switching like you're doing now). This would have the added advantage of showing the relative magnitudes of the differences in the different fluxes. Also, Qle is not defined in the text. Qle presents different behaviour than the other fluxes (Figure 2b) but this divergent result is not discussed in the text. How is this behaviour explained?

Response: We thank Dr. Lute for suggesting the figure to bring out additional details from the analysis.

We have simplified figure 2b to show the % of SDE50-days when buffer regions exhibit higher energy fluxes as compared to the urban. This convention is followed for all UBPs irrespective of positive/negative daSP.

Per reviewer's suggestion, a bar chart showing mean difference in each flux has been generated as well (Figure S9).

Qle is now defined in the text (line 368).

Possible reasons for contrasting behaviors shown by LWnet and Qle is mentioned in the manuscript (Lines 151-161).

R2-29: Line 151- I don't think these other factors were noted above.

Response: We have edited the text and modified the section significantly (Lines 125-188)

R2-30: Line 157- Why not use energy fluxes during the events instead of for the full day? Maybe this was in the methods and I missed it.

Response: The air temperature during (or at the start of) precipitation events is not only determined by the energy fluxes during the event or at the time of analysis. Instead, the boundary layer thickness and air temperature at any time during the day (assuming local land-atmosphere feedback to be the dominant mechanism), is a result of antecedent state of the system at sunrise and then the cumulative sensible heat and moisture feedback to the atmosphere till that time (McNaughton and Spriggs, 1986). Given the site and day-time specific time-lag between energy exchange near the land surface and the temperature dynamics of the atmosphere, daily energy fluxes between urban and buffers are compared. We have edited the following text in the manuscript (line 482-485).

The evaluations are performed for all days receiving SDE50 in either urban or buffer region. As antecedent states at sunrise and cumulative Q_h to the atmosphere drives the net change in boundary layer thickness and air temperature (McNaughton et al., 1986) during a day, daily energy fluxes between urban and buffers are compared.

McNaughton, K. G., & Spriggs, T. W. (1986). A mixed-layer model for regional evaporation. *Boundary-Layer Meteorology*, 34(3), 243-262.

R2-31: Figures 2c and d- please remove grey space so that plot content can be enlarged and easier to see. Might be helpful to label the colorbars to indicate what metric they are showing. I prefer a color bar that diverges at 0 for a metric like this. As it is, the colors suggest that green and brown cells mean different things, when in reality they might both be showing positive values. Why are some heat map cells divided by a black line and others aren't? Same comments apply to Figure S9.

Response: Thanks for this note. We have made significant changes to Figures 2(b), (c), and Figure S11, in response to both the reviewer's earlier suggestions and those provided in this note. Detailed information has been provided in the manuscript (line 189-205).

R2-32: Lines 160-161- by 'difference in SP' do you mean the same daSP that's been discussed throughout the paper? If not, how is this different? What is the SP for a difference in SP? For example, if this is daSP then is the SP associated with it the urban SP or the buffer SP?

Response: 'daSP' corresponds to difference between average snow probabilities of **all** SDEs encountered by buffer and urban regions during 1980-2020. The procedure to calculate daSP is described in the manuscript (Lines 441-445). It is to be noted that 'daSP' is a single value for each UBP based on the SDEs encountered by them during 1980-2020.

In contrast, 'SP' is the snow probability, and can be evaluated for a given event. We have edited figures 2c and 2d to make them clearer. Simultaneously we have edited texts in Lines 189-205 for clarifications.

R2-33: Line 162- what is the difference in fractions of SDEs? What is the fraction that is referred to? This whole paragraph should be better explained in the methods. For example, the grouping into seven bins should be in the methods.

Response: We have significantly edited figures 2c and 2d to make them simpler, and have corresponding edited texts in Lines 189-205 for clarifications.

R2-34: Line 165- what are frequency-fractions? Where was this defined?

Response: We have edited text in Lines 198-203 to clarify this now.

To this end, we evaluate the ratio of SDEs with snow probability falling within a specified range to the total number of SDE50 in each urban or buffer region. Next, the difference of these ratios between buffer and urban regions of each UBP is calculated. Average of these differences over all UBPs (Figure 2d) shows that for the SP interval 90%-100%, the fractions of SDEs between buffer and urban have the higher difference.

R2-35: Lines 159-171- I read this paragraph many times and could not figure out what it said. There seem to be several metric definitions missing, or maybe they are defined

in the methods but the same terminology is not used here. I was unable to evaluate the science in this paragraph because I couldn't understand the presentation of it.

Response: As noted above, we have edited figures 2c and 2d to make them easy to comprehend. Corresponding texts have been edited in Lines 189-205 for clarifications.

R2-36: Is there an urban-buffer contrast in precipitation amount delivered by snow-dominated events as well?

Response: Figure 3a shows the difference between annual average precipitation (buffer - urban) delivered by snow-dominated events. The details of the metrics presented in figure 3 are described in the Methods (line 445-446) for further clarification.

The second metric quantifies the difference (buffer - urban) in the annual average amount of precipitation delivered by SDEs during the analysis period, i.e., 1980-2020.

R2-37: Line 174- delete 'as well'

Response: The text has been modified in the manuscript (line 207-208).

R2-38: Line 175- I would rephrase the first few sentences of this paragraph to say that there is little contrast in a) the amount of precipitation delivered or b) the precipitation fraction between urban and buffer, instead of focusing on the relatively small differences you found.

Response: Thanks for this suggestion. We have modified the text as follows (line 209-213)

Although, an overwhelming fraction of UBPs reveal higher SP in buffer regions, only 54% of the buffer regions, out of 4,856 UBPs, receive a higher amount of annual average precipitation delivered by SDE50 (Figure 3a). This percentage reduces to 52% when urban and buffer regions are compared on the basis of annual average percentage of precipitation received through SDE50 (Figure 3(b)).

R2-39: Figure 3b- if this is a difference between two fractions that each range from 0 to 1, how can there be values greater than 1 or less than -1?

Response: Figure 3b shows the difference (buffer-urban) in annual percentage of SDE50 precipitation per total precipitation (%). Hence the values can fall within the range [100, -100]. We have edited the x-label of the figure and the caption to make this clearer. Also, details of the metrics presented in figure 3 are described in the Methods (line 440-453) for further clarification.

R2-40: Figure 3- again the grid lines are inconsistent. Not a huge deal, just annoying. In the caption, SD50 should be changed to SDE50 and either iSDE needs to be changed to match the abbreviation in the methods section (ipptSDEs) or vice versa.

Response: The abbreviations pertaining to figure 3 have been removed from the text for easier comprehension, and the caption has been modified accordingly. Also, the figure has been edited to remove the inconsistent grid lines, which were a result of pasting a high-resolution figure on the standard manuscript sheet.

R2-41: Line 187- was there any explanation for the difference in intensity offered by this previous work?

Response: ‘intensification of precipitation due to urbanization’ can be due to varied factors, which are discussed in the cited reference. For example, Li et al. (2020) notes that the mechanism involves 1) heating of air above urban surface which starts rising 2) convergence of surrounding air and continuous circulation 3) release of latent heat by rising air 4) condensation of air at the center of the city. This makes more water available for precipitation and generates an environment that favors more intense rainfall.

Overall, urbanization leads to significant increase in spatial variability of monsoon rainfall within the city and this is due to the generation or reorganization of instabilities at a local scale (~10 km). The gradients in these instabilities become the locales for intensifying rainfall patterns and lead to extreme precipitation at few urban pockets. Given that this is a bit tangential to the narrative, we have not included this in the manuscript.

Li, Y., Fowler, H.J., Argüeso, D., Blenkinsop, S., Evans, J.P., Lenderink, G., Yan, X., Guerreiro, S.B., Lewis, E. and Li, X.F., 2020. Strong intensification of hourly rainfall extremes by urbanization. *Geophysical Research Letters*, 47(14), p.e2020GL088758.

R2-42: Differences in temporal trends of snow-dominated events between urban and buffer regions, and their controls

Line 192- delete both instances of 'or'

Response: Thanks for pointing this out. We have modified the text in the manuscript (line 225-226).

R2-43: Lines 195-197- This sentence is mostly talking about afSDE and Figure 4a, but then it says 'both afSDE and aaSPSDE decrease with time in these UBPs'. This sentence makes sense with Figure 4a, but contradicts Figure 4b.

Response: We have modified the section to get rid of abbreviations and addressed this comment (please see lines 223-262).

R2-44: Figure S11- The clustering in these plots seems very strange. Is this because only statistically significant trends are shown? If so, the fact that only these trends are shown should be noted in the caption. Even if that is the case, I don't understand why there would be clusters at the 0 temperature trend lines. Trend units should be noted on x and y axis labels or in caption.

Response: Thanks for pointing this out. The earlier plot (individual) showed variations in trends (statistically significant (s.s.)) of annual frequency of snow-dominated events (on Y axis) for urban/buffer region with respect to trends of annual average temperature for the same regions. Where trends were not s.s. they were replaced with zero and that is what caused the confusion.

However, we have removed the figure and the statistics have been now presented in table S4 for better clarity.

R2-45: Figure 4- the description for the smallest semicircle in a) and b) is unclear. What does 'relatively rapid' mean? How about something like 'trend in urban region is more negative than buffer region' or 'urban trend is more negative than buffer trend'? Same applies to Figure S12.

Response: We have modified the text in the figure and its caption. Following terminology (for example) is adopted to convey that trend in urban region is more negative than the buffer (lines 229-231).

Notably, in ~57% of UBPs with s.s. negative trend (1,991 out of 3,500), urban regions show a more pronounced negative trend in annual frequency of SDE50 compared to buffer.

R2-46: Can you comment on the spatial distribution of the s.s. trends and of the locations where urban trends are greater than buffer trends?

Response: Per reviewer's suggestion, we have now carried out additional analysis. Spatial distribution of trends in the frequency of SDEs and annual average SP for UBPs where urban region show more intense negative trend as compared to the buffer are shown in Figure S14. Possible controls and their influences on these trends are as follows (Lines 232-250)

While these UBPs are distributed all across the CONUS, those with largest discrepancy in trends between urban and buffers either lie in the northernmost states east of the Mississippi River or are nestled in the mountainous regions (e.g., the Rockies of UT and CO, or the Sierras of California) of the western US (Figure S14). The more pronounced negative trend in SDEs frequency in urban areas, compared to buffers, may result from two factors: 1) a higher temperature increase in urban regions, which can potentially transition SDEs into rain events, reducing their frequency, and 2) dwindled precipitation events in urban areas, which include a reduction in SDE frequency as well. To validate these possibilities and identify the dominating factor affecting the evolution of annual frequency of SDEs, trends of mean annual temperature and annual frequency of precipitation events are obtained for the UBPs with urban regions showing pronounced negative trends (Table S4, Part A and B). Among the 1,991 UBPs where annual frequency of SDE50 exhibits a more negative trend than buffer, a noteworthy 96% of them (1908 out of 1991) show statistically significant negative trends for annual frequency of precipitation in both urban and buffer regions, with the urban regions showing a more substantial negative trend (Table S4, Part A). In contrast, only 40% of the 1,991 UBPs have higher magnitude of annual average temperature trends for urban regions. A similar pattern is observed while examining trends for SDE75 (Table S4, Part B). These results indicate that for urban regions with higher negative trend of SDEs frequency, the primary factor for the expressed contrasts is the disparity in the trends of annual precipitation frequency, rather than the air temperature.

Additional details are presented in Lines 255-262.

Largest differences in trend magnitude are observed in central and northwestern US (Figure S14 (c) and (d)). Next, we assess the potential impacts of differing trends in air temperature and critical event temperature, the two variables that show a significant association with the difference in average snow probability between urban and buffer regions (Figures 2a and S8), on the varied trends in average SP. The analysis reveals that, more so than the trend in annual average temperature, higher positive trends in annual

average event critical temperature are closely associated with a more rapid decline in the annual average SP of SDEs (Table S4, Part C and D).

R2-47: Conclusions

Line 216- should be 'analysis' not 'analyses'

Response: We have modified the text in the manuscript (line 265-266). Thanks for pointing this out.

R2-48: Line 218- rephrase as 'urbanization decreases snow probability'. 'negative impact' sounds like a value judgement

Response: We have modified the statement and removed 'negative' from the text (line 266-268).

The results indicate that in a significant fraction of UBPs, urbanization has led to an overall reduction in the frequency of snow dominated events and also the probability of snowfall during these events.

R2-49: Line 218- 'Alarmingly' seems inappropriate here without being explicit about why this should be alarming. Also, why would this be? Is it because of interactions between urbanization and changing climate, because characteristics of urban areas are changing, or something else?

Response: The word 'Alarmingly' is replaced by 'Moreover' in line 268.

Moreover, the reduced frequency of snow dominated events in urban areas is observed to intensify over time for a majority fraction of UBPs.

The impact of reduction in SDEs in urban areas are described in the lines 34-47.

R2-50: Lines 224-239- I understand the issues of scope and data limitations, but it seems like one could comment more specifically on potential contributions to urban-buffer

differences. For example, it seems likely that buffer regions have more vegetation and soil surfaces as opposed to concrete/asphalt surfaces and this could affect local convection, moisture availability, and latent heat flux in specific ways. What are the implications of less snow in urban areas? In many ways this seems like a good thing- it would likely contribute to less congestion, fewer accidents, and less funding needed to be spent on road clearing and repairs. On the other hand, if this precipitation is coming as rain instead perhaps it increases flood risk and poses challenges to drainage systems in urban areas which generally have surfaces with lower permeability.

Response: Thanks for this comment. The confounding factors affecting urban-buffer contrast are discussed partly in “Factors controlling spatial distribution and magnitude of urban-buffer snow probability contrast’ section (lines 125-188) and partly in conclusion (278-285).

We thank the reviewer for providing pointers regarding the implications of urbanization on precipitation phase. We have added the relevant text 1) describing implications of less snow on the cities in the manuscript (line 34-38) and 2) importance of considering impacts of urbanization while evaluating alterations in precipitation phase (40-47)

R2-51: Why do buffers tend to get more frequent but less intense SDE's? What are potential explanations for this? This seems like one of the more interesting findings to me.

Response: Buffer regions are relatively cooler compared to urban regions (Figs. 2a, S8a&b), so they are more likely to experience a higher probability of snow precipitation events. Therefore, even if the number of precipitation events is the same in buffer and urban areas, buffer regions are likely to encounter more frequent SDEs. Since the number of precipitation events is generally lower (with higher intensity) in urban areas, this further makes the frequency of SDEs in urban (buffer) regions to be relatively lower (higher).

There are several potential factors that contribute to a higher intensity of precipitation events in urban regions, including a higher overall temperature which enables the air to retain more moisture (Shepherd et al., 2005). The reorganization of instabilities that intensify precipitation (Li et al., 2020) and the influence of aerosols (Shepherd et al., 2005), among other factors, may also play a role. Identifying the specific causes in different UBPs is beyond the scope of this study.

Li, Y., Fowler, H.J., Argüeso, D., Blenkinsop, S., Evans, J.P., Lenderink, G., Yan, X., Guerreiro, S.B., Lewis, E. and Li, X.F., 2020. Strong intensification of hourly rainfall extremes by urbanization. *Geophysical Research Letters*, 47(14), p.e2020GL088758.

R2-52: How do you explain large negative daSP values in the West? Is there something unique about these locations?

Response: Thanks for pointing this out. Explicit analysis of the causes of relative magnitude of contrasts vis-a-vis land cover property and regional climatology has not been performed here. Such a task is challenging currently, if not insurmountable, given the lack of long-term but fine enough spatio-temporal resolution data of land and snow cover (and their properties). One potential strategy to use a continental-scale coupled land-atmosphere simulations at appropriately fine enough spatio-temporal resolution. Performing such simulations is expected to be exacting. Following text has been added to the manuscript to recognize the limitations (lines 299-307)

Complexity in analysis emanating from transient dynamics and heterogeneity of snow and land cover, in both urban and buffer regions, is another major limitation of this study. For example, removal of snow, alteration of land cover properties, heterogeneity introduced by faster or slower snowmelt in sheltered vs. exposed areas, and spatial variation of surface temperature, roughness properties, wind fields, etc., within both buffer and urban regions, can all influence land-atmosphere energy exchange and consequently the characteristics of snow dominated events. As more data becomes available, future studies may consider these details to provide a more refined understanding of their impacts.

and suggest ways for moving forward (lines 315-321)

Future research that focuses on examining the impact of the extent, spatial configuration, and other properties of built structures, including their rate of change, on snow precipitation characteristics in different UBPs, is expected to enhance these assessments and mitigation designs. Additional insights may also be gained by assessing the role of specific land covers in the buffer regions on snow dynamics, and eventually on snow-probability contrasts. The acquisition of novel, high-resolution spatiotemporal data pertaining to land surface and snow cover characteristics is likely to play a pivotal role in this regard.

R2-53: Maybe also mention that this study does not assess snow on the ground or how snowfall persists on the landscape, but is focused on snowfall. A nice future work piece might be looking at complementary data sources (ground observations or radar) to see

both if observations backup what you see with modeling/reanalysis and also to consider contrasts in snow on the ground / snow persistence.

Response: Thanks for this important comment/suggestion.

The future scope of this study has been modified to include the suggested possibilities (lines 317-321).

Additional insights may also be gained by assessing the role of specific land covers in the buffer regions on snow dynamics, and eventually on snow-probability contrasts. The acquisition of novel, high-resolution spatiotemporal data pertaining to land surface and snow cover characteristics is likely to play a pivotal role in this regard.

Methods

Data of UBPs and climate

R2-54: Figure S1- why are there extra curvilinear lines in the rasterized buffer example? Urban legend label needs to be adjusted to be next to the legend key. The map in a) could use a country outline since the color for the country and the plot background are the same. In b) it looks like there are some exceptions to the 10km buffer, where areas within 10km are excluded. Please explain these in the methods. A quick glance suggests that the data used for the energy fluxes also contains hourly temperature and precipitation. Why are these not used instead of the NLDAS data? Also, why use shortwave and net instead of shortwave and longwave?

Response: Thanks for pointing these out. Country outline has now been added to the map (Figure S1).

The shapefile of the sample UBP selected for the illustration of methodology in the last version had a waterbody passing through it (curvilinear lines). To avoid confusion, we have modified figure S1.

For some UBPs, the surrounding 10 km buffer region may overlap smaller urban regions. As noted in the text, only those grids that have more than 50% area in buffer are selected (lines 393-395). Therefore, the buffer grids that were excluded because of urban overlap appeared as exceptions.

The data used for energy flux calculation is generated using NoahMP land surface model simulations that used NLDAS climate data forcing. We have used the same dataset for our analysis all throughout the paper.

We have modified figure 2b to show the percentage of SDE50-days for which net longwave radiation fluxes (LWnet) are higher for buffer regions. Considering the dominance of SWnet as compared to LWnet in terms of magnitude in the net radiation, previously we showed only the statistics pertaining to SWnet and net radiation.

Derivation of event snow probability (SP) for UBPs

R2-55: Downscaling of temperature (and energy fluxes) to 4km using an interpolation approach seems like it would not capture urban/buffer contrasts well at all, especially since more than 80% of urban areas considered have areas smaller than the NLDAS grid.

Response: Thanks for this perceptive comment. Per reviewer's suggestion, we have now evaluated the downscaled product by comparing its representation of the urban-buffer temperature contrast with an alternative, higher spatial resolution (4 km x 4 km) PRISM product (please see the comparison in response to R2-2). This comparison demonstrates a strong alignment in the temperature contrasts between the data utilized in this analysis, which, by the way, is at subdaily resolution—ideal for event-based assessments of snow probability and other related characteristics—with the daily PRISM data.

R2-56: Line 297- the citation referenced here used inverse distance weighting to get a grid from station observations to compare to a downscaled temperature dataset, the approach was not used to downscale temperature per se. Perhaps there is a better citation you can use to justify your approach.

How is an event defined? Is it one three-hour time period with snow? This is important to note since in other works snow events have been defined as three days, or other lengths of time.

Response: In the cited reference, the bilinear interpolation approach is employed. Therein it is used to merge spatially high-resolution PRISM data and temporally high resolution NLDAS data. Hence, we have retained the reference accordingly.

Regarding the definition of 'an event' in our current analyses, it corresponds to a precipitation event. An event is defined as an instance within the temporally upscaled (i.e., 3-hour resolution) NLDAS precipitation time series when precipitation exceeds 0 mm. If, during the same period, the temperature is less than 1.146°C (corresponding to a snow probability greater than 50%), the event is classified as a snow-dominated event with SP > 50%, referred to as SDE50.

R2-57: Metrics and thresholds to assess the differences in snow precipitation characteristics between urban and buffer regions

Line 346- could rephrase as 'events for which the SP is ≥ 0.5 '

Response: Thanks for this note. We have modified the text in the manuscript (line 430-432) as follows. Since, SP is expressed in (%) in Dai's formula, 0.5 is replaced by 50%.

As the premise of this study is to evaluate the impacts of urbanization on snow events, here we focus on SDEs, i.e., events for which the snow probability $\geq 50\%$, i.e., the probability of event being in solid phase is higher or equal to than it being in the liquid phase.

R2-58: Lines 352-353- Do you mean the snow probabilities correspond to these temperatures using the Dai method? Because for the Jennings approach SP would be a function of RH as well and you wouldn't be able to define a single temperature for each SP. Please clarify in the text. Also the caption of Figure S3 should note that this is based on the Dai formula.

Response: The snow probabilities corresponding to the threshold temperatures are obtained using Dai's formula. We have modified the text in the manuscript with this detail as follows (lines 438-439).

These snow probabilities correspond to air temperature (T_{thresh}) = 1.146°C and 0.341°C, respectively (Figure S3), derived based on 'Dai' formula.

Caption for figure S3 has been also modified as per the comment.

R2-59: Line 354- please include units for each metric

Response: We have added units to every axis label in each figure. In text, we have explicitly mentioned the unit for specific metric e.g. annual average intensity of SDEs (Lines 448-451).

The fourth metric compares the intensities of precipitation (= precipitation amount / number of snow-dominated events) delivered by snow-dominated events. It is quantified in mm/events/year by subtracting the average annual intensity of SDEs over the urban region from that in the buffer.

R2-60: Line 356- please add (buffer – urban) or (urban – buffer) to clarify how daSP is calculated.

Response: We have now clarified the details pertaining to the calculation of daSP in the manuscript (line 441-445).

The first metric used here is the difference (buffer – urban) in average snow probabilities for snow dominated events, henceforth called the daSP, between the buffer and the urban regions during 1980-2020 (Figure S1 (d)). This metric helps identify whether the buffer or the urban region receive SDEs with a higher average probability of snow.

R2-61: Line 360- for pptSDE is this the average precipitation of an event or the annual sum of precipitation from these events?

Response: We have removed the abbreviations to avoid confusion. However, details about the metric are provided in the manuscript (Lines 445-446).

The second metric quantifies the difference (buffer – urban) in the annual average amount of precipitation delivered by SDEs during the analysis period, i.e., 1980-2020.

R2-62: Line 362- how is the intensity of SDE defined?

Response: Following description of intensity of SDE calculation is added in the manuscript (line 448-451).

The fourth metric compares the intensities of precipitation (= precipitation amount / number of snow-dominated events) delivered by snow-dominated events. It is quantified in mm/events/year by subtracting the average annual intensity of SDEs over the urban region from that in the buffer.

R2-63: Line 364-367- why are the trends evaluated differently? Why not use the same method for both metrics? For the trend metrics are the annual frequency and annual average SP summed across the urban and buffer components of each UBP? Or averaged? The description of the average critical temperature makes no sense to me. Why are temperature values averaged if they are below 1°C? Are these temperature values at any point in the year or just during SDEs? Why do temperature values below -4°C need to be reset to -4°C? How is this a critical temperature, what does it tell you? Maybe I'm missing something here because this is very unclear to me.

Line 374- is this 5 between the urban and buffer areas, or 5 each?

Response: Thanks for this extremely important comment. We have recalculated all the trends and carried out the entire trend analysis using non-parametric Sen's slope estimator for consistency. Trends for annual frequency of SDEs and annual average SP of SDEs are calculated for each urban and buffer region. Details of trend calculations are provided in the manuscript (Lines 451-453).

The final two metrics compare the temporal trends of 1) annual frequency of SDEs and 2) annual average SP of SDEs (for both SDE50 and SDE75) for each UBP. The trends are calculated using the non-parametric 'Sen slope' estimator.

Regarding mean critical temperature and mean critical event temperature, the details about the approach used, calculations, and threshold selection are provided in the section **Mean critical temperature** (line 454-464).

An UPB is selected if each of the regions encounter at least five events (five by urban and five by buffer). This detail is added in the manuscript (line 467-468).

Out of 7,415 UBPs (see Figure S2 (b)), 4,856 (4,553) UBPs encounter at least 5 SDE50 (SDE75) per year (Figure S2 (c, d)) in both urban and buffer regions of a UBP.

R2-64: Assessment of differences in land-atmosphere energy fluxes between urban and buffer regions.

Figure S4- Suggestion: I prefer when the color ranges are symmetric around 0. I think the text 'UBPs with positive daSP: ~91%' belongs in the manuscript text, not embedded in the figure. Histogram needs a y axis label. I would also add state outlines below the points to help the viewer better understand the spatial distribution of the points. And you might remove the box around the histogram plot. All the same comments apply to Figure 1, Figure S5, and Figure S6 as well.

Response: All the figures have been modified as suggested. Thank you for this suggestion.

R2-65: Line 386- capitalize figure

Response: We have modified the text in the manuscript (Line 479).

see supplementary Figure S4

R2-66: Line 389- so this analysis uses days of SDEs, not strictly the 3-hour time periods when they occur? Why not be more specific about the time?

Response: We have specified the reason behind the analysis being carried out for SDE50-days and not when the events occur in the manuscript as follows (line 482-485).

The evaluations are performed for all days receiving SDE50 in either urban or buffer region. As antecedent states at sunrise and cumulative Q_h to the atmosphere drives the net change in boundary layer thickness and air temperature (McNaughton and Spriggs, 1986) during a day, daily energy fluxes between urban and buffers are compared.

McNaughton, K. G., & Spriggs, T. W. (1986). A mixed-layer model for regional evaporation. *Boundary-Layer Meteorology*, 34(3), 243-262.

R2-67: Line 391- replace 'correspond to' with 'are defined as days' to clarify that this isn't fact, it is how you're defining them for the purpose of this study.

Response: We have modified the text in the manuscript (line 485-487).

Evaluations are also performed separately for a subset of these days when 1) the ground is snow-free and 2) ground is covered with snow, for both the urban and buffer regions.

RESPONSE TO THE THIRD REVIEWER'S COMMENTS

The analysis presented uses temperature and precipitation data from the North American Land Data Assimilation System, combined with a simple statistical model of precipitation type, to evaluate differences in the probability and quantities of snowfall in Urban versus Buffer (adjacent non-urban) areas (termed Urban-Buffer pairs). The authors find that Urban areas, owing to lower land surface albedo and associated greater quantities of net solar radiation, net radiation, and greater sensible heat flux, have lower snowfall probabilities than adjacent buffer areas. The analysis is interesting, well considered, and thorough and the contribution will be valued by many in the scientific community and by the general public. My comments are intended to address a few potential issues inherent in the analysis and text.

Response: We thank the reviewer for the encouraging words. We have addressed all the comments/suggestions by the reviewer.

R3-1: The authors rely on a model and not observations. I assume there is a good justification for this as it seems obvious to do, but why isn't there an observation-based analysis in the manuscript to complement the modeling results (e.g. using the Dai data set)? Related to this issue, the text should be adapted to insert, in all instances, the word "modeled" when referring to snowfall probabilities and other model-derived quantities.

Response: We thank the reviewer for this perceptive comment.

The dataset used in the studies such as (Dai, 2008; Jennings et al., 2018), even though point observations, are too coarse spatially to fall within urban and corresponding buffer regions. These data include ~12,000-point observations, with around 2,240 stations residing within the CONUS (Illustration 2). Of the total 7,415 UBPs, while 1,973 observation points fall within the urban regions and 246 in buffer regions, unfortunately there are **zero** UBPs with at least one station within both urban and corresponding buffer region. Therefore, the dataset cannot be used for the analysis carried out in this study.

Also, the available point observations generally suffer from a multitude of limitations, including substantial or frequent temporal data gaps, and lack of representativeness in relation to neighboring areas (Lines 292-295).

However, when contrasting these uncertainties with the limitations of point observation data, which often suffer from substantial or frequent temporal data gaps, lack of representativeness in relation to neighboring areas, and uneven spatial distribution, the NLDAS data emerges as a preferred choice for the current application.

As suggested by the reviewer, following text is added to the manuscript (Line 413).

“Unless stated otherwise, this modeled SP is used all throughout this study”

Illustration 2: Geographic locations of Operational Global Surface Observations dataset (DS464.0) used by Dai (2008), Jennings et al. (2018) for snow probability model parameter estimation.

R3-2: Throughout the paper there seems to be an over-use of abbreviations which make the paper unnecessarily challenging to read - the section starting on line 190 is a prime example.

Response: We thank the reviewer for this comment. We have removed unnecessary abbreviations especially pertaining to the metrics. However, for some of the terms that are used frequently, the abbreviations are retained.

R3-3: Greater emphasis needs to be placed on the analysis of Snow Dominated Events for situations with pre-existing snow cover versus events without pre-existing snow cover. Wouldn't the presence of snow largely negate land cover associated differences between urban versus buffer areas as everything is covered in highly reflective snow?

Response: We agree with the reviewer's comment. Urban-buffer energy contrast analysis has now been also separately conducted for both when urban OR buffer regions are

covered with snow, and when the ground is snow-free. Results are stated in the section 'Factors controlling spatial distribution and magnitude of urban-buffer snow probability contrast' (Lines 178-182) and illustrated and tabulated in Figure S10 and table S3.

R3-4: What is the impact of the background land cover type of the buffer areas? Given that albedo differences between pair members seems to play a large role in determining differences in snowfall probability, it would stand to reason that the albedo of buffer areas would be important. For example, a forest typically has a lower albedo than a grassland. In addition, a small amount of snow cover will dramatically alter the albedo of a grassland whereas it will have a more subtle impact on the albedo of a forest. These differences in buffer albedo would then have an appreciable impact on the difference between the buffer albedo and an adjacent urban area. Can this be evaluated with the data presented herein? Can the analysis be stratified to look at the differences in snowfall probability across differences in background buffer land cover type? Or at the very least, this effect seems worthy of discussion within the manuscript as the spatial patterns in daSP seem to evoke a land cover control.

Response: We appreciate the reviewer's insightful comment and fully agree with their observation.

Our analysis focuses on contrasting snow probabilities between urban and buffer areas within each UBP region. To achieve this, we assess a single representative temperature, and consequently snow probability time series for each region within each UBP. As the spatial resolution or unit of our data is an urban or a buffer region, it is not feasible to examine the specific impact of individual land cover types or their contrasts within buffer areas on snow probability. It's worth noting that the modeled evaluations of energy fluxes in buffer and urban regions explicitly account for the heterogeneity of land cover, including its influence on albedo and energy feedback. Hence, their aggregated influences are inherent in variations of daSP contrasts vis-à-vis energy flux contrasts (in Figs. 2b, S9, and S10). As daSP contrasts are derived from a single urban and buffer time series of snow probability, it is not feasible to explore the influence of albedo dynamics in specific land cover types on daSP within this context.

We have now highlighted the significance of the reviewer's suggestions in the text, as they could form the basis for an excellent follow-up study (Line 299-307).

Complexity in analysis emanating from transient dynamics and heterogeneity of snow and land cover, in both urban and buffer regions, is another major limitation of this study. For example, removal of snow, alteration of land cover properties, heterogeneity introduced by faster or slower snowmelt in sheltered vs. exposed areas, and spatial variation of surface temperature, roughness properties, wind fields, etc., within both buffer and urban regions, can all influence land-atmosphere energy exchange and consequently the characteristics of snow dominated events. As more data becomes available, future studies may consider these details to provide a more refined understanding of their impacts.

Importance of assessing the role played by land cover in influencing snow dynamics and eventually snow-probability contrast is also mentioned in the ‘Conclusion’ (lines 315-321)

Future research that focuses on examining the impact of the extent, spatial configuration, and other properties of built structures, including their rate of change, on snow precipitation characteristics in different UBPs, is expected to enhance these assessments and mitigation designs. Additional insights may also be gained by assessing the role of specific land covers in the buffer regions on snow dynamics, and eventually on snow-probability contrasts. The acquisition of novel, high-resolution spatiotemporal data pertaining to land surface and snow cover characteristics is likely to play a pivotal role in this regard.

R3-5: In the section starting on line 190 related to trends: Can you report the trend in daSP? Is it possible that if warming potentially pushes more pairs toward the rain-snow threshold temperature, might the subtle differences in temperature between pair members cause a greater decrease in snowfall probability an urban areas relative to the corresponding buffer area? I think the analysis suggests there is greater sensitivity in the urban areas but an explicit analysis of daSP trend seems warranted.

Response: We thank the reviewer for this comment. Per reviewer’s suggestion, we have made significant edits to the section “*Differences in temporal trends of snow-dominated events between urban and buffer regions, and their controls*”, to further assess the role of temperature and precipitation frequency trend on contrasts in trends of SDEs (see lines 225-262). An attribution analysis is carried out to assess the role of climate trends on annual frequency of SDEs and annual average SP by majority of the urban regions. It is observed that the former is driven by the faster declining trends in annual frequency of precipitation events in a high percentage of urban regions, while the latter is associated with the positive trend in annual average event critical temperature within a majority of the urban regions. The results are tabulated in Table S4 (Parts A, B, C, and D). The following text is added in the manuscript in this regard.

Lines (235-250)

The more pronounced negative trend in SDEs frequency in urban areas, compared to buffers, may result from two factors: 1) a higher temperature increase in urban regions, which can potentially transition SDEs into rain events, reducing their frequency, and 2) dwindled precipitation events in urban areas, which include a reduction in SDE frequency as well. To validate these possibilities and identify the dominating factor affecting the evolution of annual frequency of SDEs, trends of mean annual temperature and annual frequency of precipitation events are obtained for the UBPs with urban regions showing pronounced negative trends (Table S4, Part A and B). Among the 1,991 UBPs where annual frequency of SDE50 exhibits a more negative trend than buffer, a noteworthy 96% of them (1908 out of 1991) show statistically significant negative trends for annual frequency of precipitation in both urban and buffer regions, with the urban regions showing a more substantial negative trend (Table S4, Part A). In contrast, only 40% of the

REVIEWER COMMENTS

Reviewer #1 (Remarks to the Author):

The manuscript has been improved significantly in the current version. Specially, the authors added complementary materials and explanations to the data and methods part. Therefore, the manuscript is clear and readable. The findings are worth publishing in my understanding based on its novelty and solid analysis. However, I still have some suggestions.

- (1) The keywords should better contain the information of “urbanization”.
- (2) Numerical values should be added in the abstract to make the results more interesting.
- (3) The conclusion is not impressive enough due to its length.
- (4) Detail of text often requires attention.

Reviewer #3 (Remarks to the Author):

MAJOR COMMENT 1: Importance of the analysis.

Why is this work important? Why is it important to the public to know that urbanization impacts snowfall fraction? Reviewer 1 raised this question and I agree that the authors need to make the case as to why this work is important. Urban areas are not water sources so the snowfall/rainfall distinction is arguably not important in a water resources context. In my view, it is perhaps most important in terms of how the public experiences the change in precipitation type and the degree to which it is influenced by land cover change versus climate change.

For example, the final sentence of the abstract states:

“Results highlight a clear imprint of urbanization on precipitation phase and underscore the need to consider these influences while projecting hydro-meteorological risks.”

What do the authors mean by “projecting hydro-meteorological risks”? I don’t know what that means and it seems like an overtly vague statement. While this one sentence is a very specific issue, it is perhaps the most important sentence in the paper and begs the question as to why this work is important beyond random curiosity.

MAJOR COMMENT 2: Misleading language about what is “observed” versus “modeled”.

As with my comment on the previous review, the paper needs to be explicit that the results are based on a model and not observations. The authors responded that they have clarified this but I do not agree that it has been adequately addressed as nowhere does the word “model” or “modeled” appear within, or prior to, the text reporting the results. And in fact, the authors use the word(s) “observed” or “were observed” when reporting the results but in fact, these are “modeled” not “observed” quantities.

MAJOR COMMENT 3: Improving the robustness of the analyses by utilizing observed data.

The point I raised in the initial review about the lack of observation-based analyses has not been adequately addressed by the authors.

My initial comment was: “The authors rely on a model and not observations. I assume there is a good justification for this as it seems obvious to do, but why isn't there an observation-based analysis in the manuscript to complement the modeling results (e.g. using the Dai data set)? Related to this issue, the text should be adapted to insert, in all instances, the word "modeled" when referring to snowfall probabilities and other model-derived quantities.”

The authors responded as follows:

“Of the total 7,415 UBPs, while 1,973 observation points fall within the urban regions and 246 in buffer regions, unfortunately there are zero UBPs with at least one station within both urban and corresponding buffer region. Therefore, the dataset cannot be used for the analysis carried out in this study.”

RESPONSE: I do not agree with the authors on two fronts:

1) The definition of UBP constructed by the authors is somewhat arbitrary. Couldn't they simply start with the 246 observations they have identified as being in a “buffer” region and then investigate these 246 observations in comparison to the nearest urban stations? Or alternatively, they could, without too much difficulty, construct a meaningful set of criteria to identify a robust set of paired data and / or clusters of data that could be compared.

2) The authors could easily use the observations to evaluate the model estimates of snowfall probability. This would not require a paired analysis but rather would simply be aimed at evaluating the model-generated data they rely on in the paper.

MAJOR COMMENT 4: Impact of background land cover type in buffer area.

I find the authors response to my comment about the impact of buffer land cover type to be inadequate.

My initial comment was:

“What is the impact of the background land cover type of the buffer areas? Given that albedo differences between pair members seems to play a large role in determining differences in snowfall probability, it would stand to reason that the albedo of buffer areas would be important. For example, a forest typically has a lower albedo than a grassland. In addition, a small amount of snow cover will dramatically alter the albedo grassland whereas it will have a more subtle impact on the albedo of a forest. These differences in buffer albedo would then have an appreciable impact on the difference between the buffer albedo and an adjacent urban area. Can this be evaluated with the data presented herein? Can the analysis be stratified to look at the differences in snowfall probability across differences in background buffer land cover type? Or at the very least, this effect seems worthy of discussion within the manuscript as the spatial patterns in daSP seem to evoke a land cover control.”

The authors responded with:

“Our analysis focuses on contrasting snow probabilities between urban and buffer areas within each UBP region. To achieve this, we assess a single representative temperature, and consequently snow probability time series for each region within each UBP. As the spatial resolution or unit of our data is an urban or a buffer region, it is not feasible to examine the specific impact of individual land cover types or their contrasts within buffer areas on snow probability. It's worth noting that the modeled evaluations of energy fluxes in buffer and urban regions explicitly account for the heterogeneity of land cover, including its influence on albedo and energy feedback. Hence, their aggregated influences are inherent in

variations of daSP contrasts vis-à-vis energy flux contrasts (in Figs. 2b, S9, and S10). As daSP contrasts are derived from a single urban and buffer time series of snow probability, it is not feasible to explore the influence of albedo dynamics in specific land cover types on daSP within this context.”

RESPONSE: I do not agree with the authors. The authors indicate that: “As the spatial resolution or unit of our data is an urban or a buffer region, it is not feasible to examine the specific impact of individual land cover types or their contrasts within buffer areas on snow probability.” This is not factually accurate. The authors have Urban / Buffer pairs derived from model-generated data that have a resolution that is coarser than readily available land cover data products (for example the USGS National Land Cover Data Set is available – I believe – at 30 m resolution). The resolution of this land cover data set would be entirely sufficient for the authors to follow my suggestion and to analyze their results in a stratified manner based on predominant buffer land cover type. The background albedo of, for example, a buffer that is dominated by forest versus grassland would most certainly impact the albedo differences between buffer and urban pairs, both without and with pre-existing snow cover. I won’t restate my initial point but I encourage the editor and authors to re-read my initial point carefully as first principles suggest the background land cover type would be very important in terms of how the albedo changes with and without snow cover. Moreover, the authors indicate that the models used in their analysis honors the land cover type of the areas analyzed within the land-atmosphere energy exchange calculation and while that is certainly true, that is not relevant to the point made in my initial comment.

Smaller Comment: I am confused by this sentence, starting on line 57:

“In fact, the Global Urban Heat Island Data Set, v1 (2013)38, which consists of data pertaining to differences in average day time maximum and night time minimum temperature during summer months of 2013 for 31,500 urban and 60 corresponding buffer regions reveal that 8,949 (28%) urban regions during day time and 11,514 61 (36%) regions during night are cooler than their buffer counterparts.” Is this an accidental mis-statement? The statement indicates that urban areas are “cooler” than buffer counter-parts? That seems opposite of the premise of the entire paper and opposite to intuition. If this is indeed a correct statement, then perhaps the authors could state that this is counter-intuitive?

Reviewer #1 (Remarks to the Author):

The manuscript has been improved significantly in the current version. Specially, the authors added complementary materials and explanations to the data and methods part. Therefore, the manuscript is clear and readable. The findings are worth publishing in my understanding based on its novelty and solid analysis. However, I still have some suggestions.

We appreciate the reviewer's favorable comments and for acknowledgement of the novelty and potential impact of this work. We have incorporated the suggested changes. More details are presented below.

(R1-1) The keywords should better contain the information of "urbanization".

We thank the reviewer for pointing this out. The keyword list has now been edited to include 'Urbanization impacts'.

(R1-2) Numerical values should be added in the abstract to make the results more interesting.

In the last round of review (comment number R2-8 in the first review), it was suggested to remove the numerical values. Per this reviewer's suggestion and mindful of the comments that we got in the last round of review, we have added numbers pertaining to a few critical results (see Lines 18-21).

Among 4,856 urban-buffer pairs that received at least five SDEs per year, a majority of urban regions are characterized by smaller snow probability (81%), lower frequency (99%), and faster declining trends in SDEs (57%), compared to their buffer counterparts.

(R1-3) The conclusion is not impressive enough due to its length.

We thank the reviewer for this comment. To ensure that the Conclusion is emphatic, we have moved some text that noted that limitations from the "Conclusions" section into a newly created "Limitations" section (see Lines 279-324).

(R1-4) Detail of text often requires attention.

We thank the reviewer for the note. We have again gone through the text thoroughly and made sure that the typographical and grammatical errors are addressed.

Reviewer #3 (Remarks to the Author):

MAJOR COMMENT 1: Importance of analysis.

Why is this work important? Why is it important to the public to know that urbanization impacts snowfall fraction? Reviewer 1 raised this question and I agree that the authors need to make the case as to why this work is important. Urban areas are not water sources so the snowfall/rainfall distinction is arguably not important in a water resources context. In my view, it is perhaps most important in terms of how the public experiences the change in precipitation type and the degree to which it is influenced by land cover change versus climate change.

For example, the final sentence of the abstract states:

“Results highlight a clear imprint of urbanization on precipitation phase and underscore the need to consider these influences while projecting hydro-meteorological risks.”

What do the authors mean by “projecting hydro-meteorological risks”? I don’t know what that means and it seems like an overtly vague statement. While this one sentence is a very specific issue, it is perhaps the most important sentence in the paper and begs the question as to why this work is important beyond random curiosity.

We thank the reviewer for this comment. In response to Reviewer 1’s question in the last round of review, we had modified the text in the main manuscript. The revised text (Lines 34-40) also addresses this reviewer’s suggestion to better highlight the importance/relevance of this work. Excerpts are provided below.

“Especially within the urban regions, the phase of precipitation also influences traffic mobility²⁰ and road accident risks^{21,22}, frequency and intensity of deicing salt application and its consequent impacts on water quality²³, contamination from runoff^{24,25}, and vulnerability from extreme hydro-meteorological events²⁶. While the influence of climate variations and changes on the likelihood of snow vs. rain has been studied extensively^{2,3,27,28}, whether, to what extent, and where urbanization impacts precipitation phase over the continental USA (CONUS) remains unknown.”

By the statement, “Results highlight a clear imprint of urbanization on precipitation phase and underscore the need to consider these influences while projecting hydro-meteorological risks.” (lines 24-25), we mean “an exclusive focus on evaluating alterations in precipitation phase solely due to global warming, without considering the potential regional impacts of urbanization, would lead to deficient or ineffective projections of its consequences and subsequent mitigation strategies.” This has been noted in lines 40-43.

Since, urbanization is indeed found to influence the precipitation phase, we recapitulate the importance of understanding these influences while projecting hydro-meteorological risks in the conclusion (lines 346-352).

“Considering that alterations in snow vs. rain precipitation can have wide-ranging impacts, such as on water supply¹, flood risk⁶ and its quantification²⁶, traffic operations

and safety^{21,22}, water quality²³, and drought and its prediction⁴⁸, the results of this study emphasize the importance of considering the influence of urbanization on snow precipitation characteristics while projecting influences on the aforementioned variables. Importantly, failing to consider the potential influences of urbanization on the characteristics of snow precipitation could lead to incomplete or ineffective projections of the hydro-meteorological risks it presents.”

MAJOR COMMENT 2: Misleading language about what is “observed” versus “modeled”.

As with my comment on the previous review, the paper needs to be explicit that the results are based on a model and not observations. The authors responded that they have clarified this but I do not agree that it has been adequately addressed as nowhere does the word “model” or “modeled” appear within, or prior to, the text reporting the results. And in fact, the authors use the word(s) “observed” or “were observed” when reporting the results but in fact, these are “modeled” not “observed” quantities.

Right at the beginning of the ‘Results’ section, which starts at line 81, we explicitly mentioned “*Modelled snow probability*”. We then pointed the readers to Methods in line 83, wherein it was explicitly mentioned (now removed in the latest version) in lines 413-414 (**line numbers 413-414 are based on the manuscript version uploaded after the first round of review**) that “*Unless stated otherwise, this modeled SP is used all throughout this study.*”

To make it clearer, we have now added the following text right before the results in Lines 73-75

“Unless stated otherwise, reported snow probabilities, their contrasts, and subsequent analysis all throughout this study are based on modeled estimates of likelihood of snow precipitation.”

In addition, in lines 65-67, we have edited the statements to “*This study assesses the differences in **modeled** average snow probability between urban areas and their buffer regions, henceforth termed urban-buffer pairs or UBPs, throughout the CONUS.*” .

Furthermore, edits have also been made in the Conclusions. See lines 328-330.

*“The results indicate that in a significant fraction of UBPs, urbanization has led to an overall reduction in the frequency of snow dominated events and also the **modeled** probability of snowfall during these events.”*

There were 12 instances of the word “observed” in the main manuscript, and in all instances “observed” was used to indicate authors’ “observations” of the results. To avoid confusion, we have replaced or removed the term “observed” in Lines 85-87, 88-89, 90-93, 101-102, 119-121, 137-140, 148-149, 262, 328-330, 330-332, 336-337, 389-390. The only instance of the word ‘observed’ is on the line 302 which refers to the results obtained from observations.

MAJOR COMMENT 3: Improving the robustness of the analyses by utilizing observed data.

The point I raised in the initial review about the lack of observation-based analyses has not been adequately addressed by the authors.

My initial comment was: “The authors rely on a model and not observations. I assume there is a good justification for this as it seems obvious to do, but why isn't there an observation-based analysis in the manuscript to complement the modeling results (e.g. using the Dai data set)? Related to this issue, the text should be adapted to insert, in all instances, the word "modeled" when referring to snowfall probabilities and other model-derived quantities.”

The authors responded as follows:

“Of the total 7,415 UBPs, while 1,973 observation points fall within the urban regions and 246 in buffer regions, unfortunately there are zero UBPs with at least one station within both urban and corresponding buffer region. Therefore, the dataset cannot be used for the analysis carried out in this study.”

RESPONSE: I do not agree with the authors on two fronts:

1) The definition of UBP constructed by the authors is somewhat arbitrary. Couldn't they simply start with the 246 observations they have identified as being in a “buffer” region and then investigate these 246 observations in comparison to the nearest urban stations? Or alternatively, they could, without too much difficulty, construct a meaningful set of criteria to identify a robust set of paired data and / or clusters of data that could be compared.

2) The authors could easily use the observations to evaluate the model estimates of snowfall probability. This would not require a paired analysis but rather would simply be aimed at evaluating the model-generated data they rely on in the paper.

We thank the reviewer for further clarifying the point and providing possible ways to carry out the analysis. **After careful deliberation, we have conducted additional analysis based on reviewer's suggestion in point 2.** A detailed response for each point is provided below.

Regarding reviewer's point 1: It is to be noted that the selection criteria for the buffer region corresponding to each urban region (i.e., the definition of UBP) are derived from the literature. The underlying thesis is that the neighborhood buffer of an urban region is expected to have similar climatology due to its proximity to the urban region, and the differences in studied climatic characteristics are likely due to contrasts in land cover (and consequently land-atmosphere feedback) between the two regions. In addition to the logistical challenge of defining new UBPs that may not adhere to the aforementioned thesis, using station-level data for defining UBPs poses additional challenges:

- a) There are only 1,137 stations which reside either in urban or buffer regions. In contrast, the NLDAS data allows analyses in 4,856 UBPs.
- b) The station data is not exhaustive temporally. 67% of the stations, i.e., 770 out of 1,137, have data with a duration of fewer than 10 years. In addition, they lack spatiotemporal consistency, i.e., different stations have datasets with different durations, and start and end times. These variations in the length and the time period of the data make it difficult to compare snow probability (and its trend) contrasts across sites. For example, if there are two urban regions with 5 years and 25 years of data, respectively, the trends calculated for both regions cannot be compared as they are for different periods. Furthermore, the statistical properties of short-term data are likely to vary from the long-term period of our analysis. These limitations constrain the applicability of station data for this study.

The following lines are added to the manuscript (Lines 283-296).

“Alternative is to use the station-level data of occurrence of snow vs. rain that have been used to develop snow probability models^{45,46}. However, these point observation data suffer from several limitations, including 1) lack of sufficient stations to adequately perform a continental analysis over the selected UBPs, 2) lack of cotemporaneous, long-term data (e.g., 67% of stations residing within UBPs have data for less than 10 years) even in the UBPs that have data of precipitation phase in either urban or the buffer region, 3) lack of representativeness to the neighboring areas. Considering these limitations, the NLDAS data emerges as a preferred choice for the current application. It is to be acknowledged that coarse NLDAS data (spatial resolution of $0.125^\circ \times 0.125^\circ$) of albedo and land-atmosphere energy exchanges that are sourced from NLDAS project make a comprehensive attribution analysis of snow probability contrasts a challenge. Although albedo data can be obtained at higher resolutions through remote sensing, these estimates are unavailable on days with SDE50 due to cloud cover. Daily remotely sensed albedo data products for CONUS usually only offer the daily albedo estimate within a specific timeframe or window, excluding data for the cloudy days.”

Regarding point 2: We appreciate the reviewer's suggestion to validate the fidelity of our results against station-level data. To this end, we performed additional analysis in two parts. First, we compared the trends of annual event-average SP obtained with the station data and NLDAS data. This analysis is carried out by identifying the NLDAS grids close to the selected stations and calculating the trends of annual event-avg. SP. The second analysis compares the daSP of the stations lying within urban and buffer regions of an UBP. The details of both analyses are as follows.

Analysis 1: Comparison of trend between station and NLDAS data

The stations that reside within the urban/buffer regions are identified. Details about the number of stations within urban and buffer regions are provided in Table 1 below.

Table 1: Number of stations within urban and buffer regions

Step #	Quantity	Frequency
1	Total number of stations across the globe	11,924
	Stations within CONUS	2,240
2	Among the 7415 UBPs within CONUS	
	Number of stations within urban regions	1,973
	Number of stations within buffer regions	246
3	Among the 4856 UBPs selected based on number of snow event criteria	4,856
	Number of stations within urban regions	1,044
	Number of stations within buffer regions	116
	Total	1,160

Since, it is a comparative analysis of annual event-avg. SP, some stations are excluded from the analysis:

- Out of 1,160, for 12 stations, the number of snow events in the observation data is 0.
- For two stations, the station data ends before 1980. Since NLDAS data starts from 1980 and this being a comparative study, these two stations are discarded as well.
- Eight stations have data for just one day while one additional station has data for four days only. During that period, NLDAS data does not have any SDE50.

Therefore, the number of remaining stations is 1,137. Out of these, 1,022 stations reside within the urban regions and 115 reside within the buffer. For each station, data from the nearest NLDAS grid (for the same duration as the observational data at the station) is obtained, and trends in annual event-average SP are calculated using Sen's slope method. For the stations with statistically significant trends (p-value < 0.05), the directions of trends (positive/negative) are compared with those obtained from NLDAS data (see Illustration 1). Out of the 1,022 (115) stations within urban (buffer) regions, the number of stations with statistically significant trends is 110 (11).

Illustration 1: Comparison between station data and NLDAS data based on the trends of annual event-average SP.

Out of the 110 stations within urban regions, the trend directions of station and NLDAS annual event-average SP match for 70 stations (~64%) (Illustration 1a). In the case of buffer regions, the trend directions match for 100% of the stations. Overall, out of 121 stations, 81 stations (~67% stations) show trends in the same direction for both datasets (highlighted quadrants).

Analysis 2: Comparison of daSP calculated with station and NLDAS data.

We identified 29 UBPs that have at least one station in both urban and buffer regions. Due to empty station titles, these were inadvertently filtered out in response to the previous review. For these stations, nearest NLDAS grids are identified (as in the previous analysis), and data are extracted for the same duration. daSP is calculated for each UBP using both datasets and compared (see Illustration 2). Out of the 29, 17 UBPs show statistically significant daSP for station data. For 12 out of the 17 UBPs (71% of regions), the daSP signs are the same for both station and NLDAS data.

Illustration 2: Comparison between daSP calculated using station and NLDAS data. For 71% of UBPs, the contrast of SP from station data matches that from NLDAS (light blue).

Based on the two comparisons, it is evident that there is alignment between the results obtained with gridded and station data, underscoring the credibility of the analysis carried out with gridded data and implying that the findings are reliable. It is to be noted that some discrepancy is expected between NLDAS and observational data due to: 1) the scale mismatch between the two datasets—the observational data at a point may lack representativeness of the neighboring areas; 2) the uncertainties inherent in the two datasets. We have added the following in Lines 296-307.

“Notably, comparison of modeled trend of annual event average snow probability from NLDAS data with observations, at 121 locations where the modeled trends are statistically significant (p -value < 0.05), reveals that the direction of trends (i.e., positive or negative) align at 67% of the stations (Figure S15). Furthermore, among the 17 UBPs with at least one observation station in both urban and buffer regions, and where the snow probability contrast is statistically significant, 71% of the UBPs exhibit alignment between modeled and observed regarding whether the urban or the buffer region has a higher snow probability (Figure S15). These findings indicate that evaluations based on NLDAS data generally align with observations. Discrepancies in the snow probability or its contrasts based on the two data sources likely arise from 1) the scale mismatch, where observational data at a single point may lack representativeness of neighboring areas; 2) the inherent uncertainties in the two datasets.”

MAJOR COMMENT 4: Impact of background land cover type in buffer area.

I find the authors response to my comment about the impact of buffer land cover type to be inadequate.

My initial comment was:

“What is the impact of the background land cover type of the buffer areas? Given that albedo differences between pair members seems to play a large role in determining differences in snowfall probability, it would stand to reason that the albedo of buffer areas would be important. For example, a forest typically has a lower albedo than a grassland. In addition, a small amount of snow cover will dramatically alter the albedo grassland whereas it will have a more subtle impact on the albedo of a forest. These differences in buffer albedo would then have an appreciable impact on the difference between the buffer albedo and an adjacent urban area. Can this be evaluated with the data presented herein? Can the analysis be stratified to look at the differences in snowfall probability across differences in background buffer land cover type? Or at the very least, this effect seems worthy of discussion within the manuscript as the spatial patterns in daSP seem to evoke a land cover control.”

The authors responded with:

“Our analysis focuses on contrasting snow probabilities between urban and buffer areas within each UBP region. To achieve this, we assess a single representative temperature, and consequently snow probability time series for each region within each UBP. As the spatial resolution or unit of our data is an urban or a buffer region, it is not feasible to examine the specific impact of individual land cover types or their contrasts within buffer areas on snow probability. It's worth noting that the modeled evaluations of energy fluxes in buffer and urban regions explicitly account for the heterogeneity of land cover, including its influence on albedo and energy feedback. Hence, their aggregated influences are inherent in variations of daSP contrasts vis-à-vis energy flux contrasts (in Figs. 2b, S9, and S10). As daSP contrasts are derived from a single urban and buffer time series of snow probability, it is not feasible to explore the influence of albedo dynamics in specific land cover types on daSP within this context.”

RESPONSE: I do not agree with the authors. The authors indicate that: “As the spatial resolution or unit of our data is an urban or a buffer region, it is not feasible to examine the specific impact of individual land cover types or their contrasts within buffer areas on snow probability.” This is not factually accurate. The authors have Urban / Buffer pairs derived from model-generated data that have a resolution that is coarser than readily available land cover data products (for example the USGS National Land Cover Data Set is available – I believe – at 30 m resolution). The resolution of this land cover data set would be entirely sufficient for the authors to follow my suggestion and to analyze their results in a stratified manner based on predominant buffer land cover type. The background albedo of, for example, a buffer that is dominated by forest versus grassland would most certainly impact the albedo differences between buffer and urban pairs, both without and with pre-existing snow cover. I won't restate my initial point but I encourage the editor and authors to re-read my initial point carefully as first principles

suggest the background land cover type would be very important in terms of how the albedo changes with and without snow cover. Moreover, the authors indicate that the models used in their analysis honors the land cover type of the areas analyzed within the land-atmosphere energy exchange calculation and while that is certainly true, that is not relevant to the point made in my initial comment.

We thank the reviewer for providing additional details and clarifications about the comment. As we understand it, the initial comment encouraged us to investigate the variations in snowfall probability contrasts in relation to background buffer land cover types. This suggestion stems from the recognition that albedos before and after snowfall are influenced by land cover types. While we had demonstrated the contrasts of albedo between UBPs vis-à-vis their snow probability contrasts, the reviewer's comment prompts us to further quantify the imprint of land cover differences on albedo and, consequently, snow probabilities.

To address this, an ideal approach would involve utilizing a high-resolution (spatial) climate and albedo dataset, with a resolution sufficient for mapping to various land covers. However, the daily albedo data utilized in this study is sourced from NLDAS and has a resolution of approximately $\sim 0.125^\circ \times 0.125^\circ$. Unfortunately, daily remotely sensed albedo data for CONUS on days when SDE50 occurs is unavailable due to cloudiness. Typical daily albedo products only provide the best albedo estimate within a time frame, and exclude data for cloudy days. If a dataset with finer spatial resolution that provided data on cloudy days were available, it would enable the mapping of albedo data to specific land covers, allowing for a comparison of albedo contrasts within and across UBPs vis-à-vis the background land cover of buffer. Since, such a fine resolution dataset at the continental scale is unavailable at present, in the last round of review we honored the third suggestion by the reviewer i.e., **“Or at the very least, this effect seems worthy of discussion within the manuscript as the spatial patterns in daSP seem to evoke a land cover control”**. In the main manuscript, we acknowledge this as one of the limitations of the study and recognized the possibility of future scope in this regard (lines 317-324).

We do recognize the significance of the reviewer's point, and had there been albedo data with a sufficiently fine spatio-temporal resolution, we would have conducted a comprehensive attribution analysis to assess the influence of land cover on contrasts in snow probability. Given the constraints of coarse spatial data currently available, **here we conduct an alternative analysis in line with the reviewer's suggestion, to assess whether UBPs that experience positive vs. negative daSP have marked differences in land covers in their buffers**. But, before we proceed further, we would like to provide a summary of albedo related analysis described in the main manuscript for the sake of continuity of our response. Based on the statistics (% days when urban albedo is higher/lower than buffer region on SDE50 days), albedo is identified as **one of the factors** modulating energy contrast. This analysis was carried out separately for a) snow covered ground conditions, b) snow-free ground conditions, and c) either a or b conditions is satisfied. Our analysis 1) **identified albedo contrasts as one of the factors**

likely affecting energy contrast and 2) **highlighted the role of other factors (lines 186-188, 311-316) which probably also contribute to the energy and snow probability contrast.** While discussing the limitation of this study (lines 317-324), we also acknowledge that lack of data regarding transient dynamics and heterogeneity of snow and land cover, in both urban and buffer regions is a limitation of this study. It is to be noted here that albedo differences across different buffer regions, especially during the snow season, is not only dependent on the land cover. Example of this is illustrated in the figure below where we can see that the same land cover can have different albedos across different UBPs (Illustration 3, c,d). Other major factor may modulate differences in albedo including the frequency and intermittency of snow events, aerosol and soot concentrations, temperature, and solar forcings (Gardner and Sharp, 2010; Wang and Davidson, 2007). Presence of these confounding factors are likely to further interfere with a clear imprint of contrasts in background buffer land covers on contrasts in snow probability.

Illustration 3: Variation of hourly albedo during 2019 (hour zero is 12 a.m. Coordinated Universal Time (UTC) on January 1, 2019) over different land covers in four randomly selected UBPs. Plots within a panel indicate land covers within a single buffer region.

As an additional analysis, we now first select all buffer regions (out of 1,387 UBPs with statistically significant daSP) with distinct NLDAS grid(s) within both urban and buffer. Subsequently, we evaluate the predominant land cover within each of these NLDAS grids (note: the predominant land cover for each NLDAS grid was also used in the Noah snow albedo model simulations within the NLDAS project) for each buffer. The average fractional contribution of albedo from 11 NLCD land covers—water, developed areas, deciduous forest, evergreen forest, mixed forest, shrub (open), grassland, hay, cultivated crops, woody wetlands, and herbaceous wetlands—over SDE50 snow cover-free days in each buffer is then assessed. Finally, the contributions of each land cover are averaged separately over all buffers belonging to UBPs with positive and negative daSP.

Table: Fractional contribution of albedo by different land covers during SDE50, snow-free days for buffer regions belonging to positive and negative daSP.

No.	Land covers	Positive daSP	Negative daSP
1	Water	3.168	7.778
2	Developed	0.128	0.363
3	Deciduous forest	29.729	43.526
4	Evergreen forest	13.077	3.083
5	Mixed forest	2.097	20.56
6	Shrub (open)	3.766	0
7	Grassland	5.29	0
8	Hay	4.885	8.036
9	Cultivated crops	35.037	9.173
10	Woody wetlands	2.822	7.133
11	Herbaceous wetlands	0	0.348

It may be noted from the table that UBPs with positive daSP exhibit a higher proportion of albedo contributed by cultivated crops in the buffer region. Notably, cultivated crops, grasslands, and shrubs generally have higher albedos compared to built-up areas. Conversely, UBPs with negative daSP are distinguished by a higher albedo contribution from deciduous and mixed forests in their buffers—land covers characterized by albedos lower than those of built-up areas.

Following text has been edited in the manuscript:

Lines 190-203:

“To assess whether albedo differences between positive and negative daSPs on SDE50 days with snow-free ground conditions are primarily due to contrasts in land cover types in their buffers, we evaluate the average fractional contribution to albedo for 11 land covers: water, developed areas, deciduous forest, evergreen forest, mixed forest, shrub (open), grassland, hay, cultivated crops, woody wetlands, and herbaceous wetlands. This evaluation is performed separately over buffers with positive and negative daSPs. The fractional contribution to buffer albedo for a given land cover is obtained as the ratio of the area-weighted albedo of that land cover to the product of the total area of the buffer and its albedo. The ratio is then averaged over all SDE50 events with snow-free ground conditions. The results (see Table S4) indicate that UBPs with negative daSP are distinguished by a higher albedo contribution from deciduous and mixed forests in their buffers—land covers characterized by albedos lower than those of built-up areas. In contrast, UBPs with positive daSP exhibit a higher proportion of albedo contributed by cultivated crops, grasslands, and shrubs—land covers generally featuring higher albedos compared to built-up areas.”

Smaller Comment: I am confused by this sentence, starting on line 57:

“In fact, the Global Urban Heat Island Data Set, v1 (2013)38, which consists of data pertaining to differences in average day time maximum and night time minimum temperature during summer months of 2013 for 31,500 urban and 60 corresponding buffer regions reveal that 8,949 (28%) urban regions during day time and 11,514 61 (36%) regions during night are cooler than their buffer counterparts.”

Is this an accidental mis-statement? The statement indicates that urban areas are “cooler” than buffer counter-parts? That seems opposite of the premise of the entire paper and opposite to intuition. If this is indeed a correct statement, then perhaps the authors could state that this is counter-intuitive?

Thanks for this note. There is no mistake in this statement. As described in lines 55-57, “*Over the continent, urbanization’s impacts on precipitation phase may however experience significant spatial heterogeneity due to the fact that several locations actually experience urban cooling, instead of heating^{36,37}*”, **yes urban areas can experience urban cooling**. We further note in lines 60-61 that “*...(28%) urban regions during day time and 11,514 (36%) regions during night are cooler than their buffer counterparts*”. **These statements are used to make a case that snow precipitation contrasts with urbanization can be diverse and is nonobvious**. Furthermore, as noted in lines 61-64, “*In addition, given urbanization is also known to alter precipitation temporality and its characteristics³⁹⁻⁴¹, its eventual impact on snow frequency and amount is likely to be influenced by changes in air temperature as well as the temporal distribution and frequency of precipitation events*”

References

Gardner, A.S. and Sharp, M.J., 2010. A review of snow and ice albedo and the development of a new physically based broadband albedo parameterization. *Journal of Geophysical Research: Earth Surface*, 115(F1).

Wang, S. and Davidson, A., 2007. Impact of climate variations on surface albedo of a temperate grassland. *Agricultural and Forest Meteorology*, 142(2-4), pp.133-142.

REVIEWER COMMENTS

Reviewer #3 (Remarks to the Author):

The authors have added additional analyses (and / or text) to address my previous comments. In all but one instance, these revisions have been satisfactory. The remaining issue relates to my repeated comment in each review about the lack of analysis regarding the buffer land cover types and their impacts on the results in the context of the background albedo values that are at the heart of the papers conclusion that land cover impacts on albedo impact energy fluxes broadly, which in turn impact sensible heating to the atmosphere, and subsequently impacts the probability of snowfall (and the associated difference between snowfall probability in urban versus buffer areas).

The materials in illustration 3, presented by the authors in their response document, lacks sufficient detail to be fully used to address my comment but the data do, in fact, lend further credence to my point that background buffer land cover type matters immensely. Note that in that illustration one can see very large differences in albedo between land cover types when they are snow-free and very large delta-albedo values when snow cover is present versus when it is not. They also show that these delta albedo values are very different for the different land cover types (eg grassland versus forest) which is consistent with the point I have been making since the beginning of the review process.

In addition, upon re-reviewing the maps of daSP shown in figures S4 and S5 it is obvious that the statistically significant positive daSP values (i.e. the hallmarks of the papers premise) are in areas that (based on general knowledge of the biogeography of the US) are much more likely to be forested areas. Note in those maps that almost none of the positive (and significant) daSP values occur in the plains (i.e. grasslands) or in areas of significant agricultural activity - which are largely devoid of forests. In addition, the positive daSP's are nearly all in the mountainous west or along the mountains of the Eastern US - which are much more likely to be forested. While I cannot evaluate this hypothesis from the data presented by the authors, it is an obvious question to ask given the data presented:

What is the role of the background buffer type on the results presented in this paper?

The authors have argued that they would need high resolution albedo data to address my point but never was this suggested. Rather, I simply suggested the authors stratify their analysis of daSP (and energy fluxes) using readily and publicly available land cover data (such as the USGS National Land Cover Data Base). These data are high enough resolution and a simple GIS analysis could classify each buffer area as being dominant or majority of one land cover type or another, and / or be binned based on percentages of different land cover types. Because land cover types are relatively static, this does not need to be done in a temporally variable manner but rather would only involve the analysis of a single geographic layer of the US.

While the authors have clearly improved their paper, it is my opinion that the results presented demand that the background buffer land cover types be included as a basis for the analyses included in the paper.

The authors have added additional analyses (and / or text) to address my previous comments. In all but one instance, these revisions have been satisfactory.

Response:

We thank the reviewer for appreciating the additional analysis carried out in response to the comments.

The remaining issue relates to my repeated comment in each review about the lack of analysis regarding the buffer land cover types and their impacts on the results in the context of the background albedo values that are at the heart of the papers conclusion that land cover impacts on albedo impact energy fluxes broadly, which in turn impact sensible heating to the atmosphere, and subsequently impacts the probability of snowfall (and the associated difference between snowfall probability in urban versus buffer areas).

The materials in illustration 3, presented by the authors in their response document, lacks sufficient detail to be fully used to address my comment but the data do, in fact, lend further credence to my point that background buffer land cover type matters immensely. Note that in that illustration one can see very large differences in albedo between land cover types when they are snow-free and very large delta-albedo values when snow cover is present versus when it is not. They also show that these delta albedo values are very different for the different land cover types (eg grassland versus forest) which is consistent with the point I have been making since the beginning of the review process.

In addition, upon re-reviewing the maps of daSP shown in figures S4 and S5 it is obvious that the statistically significant positive daSP values (i.e. the hallmarks of the papers premise) are in areas that (based on general knowledge of the biogeography of the US) are much more likely to be forested areas. Note in those maps that almost none of the positive (and significant) daSP values occur in the plains (i.e. grasslands) or in areas of significant agricultural activity - which are largely devoid of forests. In addition, the positive daSP's are nearly all in the mountainous west or along the mountains of the Eastern US - which are much more likely to be forested. While I cannot evaluate this hypothesis from the data presented by the authors, it is an obvious question to ask given the data presented: What is the role of the background buffer type on the results presented in this paper?

The authors have argued that they would need high resolution albedo data to address my point but never was this suggested. Rather, I simply suggested the authors stratify their analysis of daSP (and energy fluxes) using readily and publicly available land cover data (such as the USGS National Land Cover Data Base). These data are high enough resolution and a simple GIS analysis could classify each buffer area as being dominant or majority of one land cover type or another, and / or be binned based on percentages of different land cover types. Because land cover types are relatively static, this does not need to be done in a temporally variable manner but rather would only involve the analysis of a single geographic layer of the US.

While the authors have clearly improved their paper, it is my opinion that the results presented demand that the background buffer land cover types be included as a basis for the analyses included in the paper.

Response:

We appreciate the reviewer’s emphasis to stratify daSP based on land cover to further tease out the importance of buffer land cover types on the results. **To directly align our analysis with reviewer’s suggestion**, we now map NLCD land cover to each buffer region of UBP that exhibited statistically significant daSP. The percentage area occupied by each land cover is then calculated. After segregating the buffer regions based on whether they belong to the UBP with positive/negative daSP, mean of percentage land area for each land cover is calculated. Based on results, it is observed that very much along the lines of what the reviewer astutely noted, crop fraction in positive daSP is overall much larger than in negative daSP. Also, negative daSP is characterized by higher total fraction of forests (deciduous, evergreen, and mixed). We edited the text in the main manuscript to reflect this analysis in lines 191-206.

“To assess the imprint of buffer land covers on daSP contrasts on SDE50 days with snow-free ground conditions, we stratify the aerial fraction of 11 land covers: water, developed areas, deciduous forest, evergreen forest, mixed forest, shrub (open), grassland, hay, cultivated crops, woody wetlands, and herbaceous wetlands. Fractions of each land cover are first obtained by mapping National Land Cover Database (NLCD) land cover to each buffer region with statistically significant daSP (Figure S4). The average of these fractions for each land cover is then evaluated separately over UBPs with positive and negative daSP. Results (Table S4) indicate that UBPs with negative daSP are distinguished by a higher forest coverage (deciduous, mixed, and evergreen forests). In contrast, UBPs with positive daSP have a higher proportion of cultivated crops. This imprint is also reflected in sparse occurrence of negative daSP UBPs in croplands of mid-western US. While forest (cultivated crop) land covers usually are characterized by lower (higher) albedos than in built-up areas, it is to be noted that land-atmosphere energy feedback in buffers are also determined by differences in surface and aerodynamic conductances⁴⁴ as well. Moreover, the impacts of additional land covers, though individually minor, could wield a more substantial influence than the predominant land cover in the area. Mapping the influences of each land cover on different energy components with desired certainty currently remains beyond the scope of this study.”

Table S4: Mean land cover fraction for buffer regions belonging to positive and negative daSP.

Land covers	Fraction of land cover (%)	
	Positive daSP	Negative daSP
Water	3.942	3.265
Developed	4.418	2.375
Barren	0.449	1.073
Deciduous	20.288	9.144
Evergreen	10.257	28.250
Mixed	6.682	4.185
Shrub	11.824	25.160
Grassland	5.022	11.068
Hay	8.209	3.219
Crops	22.481	9.168
Woody Wetlands	5.004	2.130
Herbaceous Wetlands	1.422	0.891

While results reveal the influence of land covers over albedo, we have also acknowledged and emphasized that not ALL reported contrasts in SP between urban and buffer is attributable to albedo differences. Contrasts in surface and aerodynamic conductance originating from both morphometric and dynamic hydraulic properties of land covers also affects air temperature and consequently SP contrasts (Lines 160-161)

REVIEWERS' COMMENTS

Reviewer #3 (Remarks to the Author):

The authors have appropriately addressed my final comment and I believe the paper can be published without further revision - however, I strongly urge the authors and the editorial staff to double check the paper for typos and / or grammatical errors (as a standard practice).

Reviewer #3 (Remarks to the Author):

The authors have appropriately addressed my final comment and I believe the paper can be published without further revision - however, I strongly urge the authors and the editorial staff to double check the paper for typos and / or grammatical errors (as a standard practice).

Response

We thank the reviewer for providing a thoughtful feedback throughout.

We have thoroughly gone through the manuscript and supplementary document to ensure absence of any typographical/grammatical errors in the final version.